# Enhancing the usability of weather radar data for the statistical analysis of extreme precipitation events

Andreas Hänsler[1], Markus Weiler[1]

[1]Chair of Hydrology, University of Freiburg, 79098 Freiburg, Germany

*Correspondence to*: Andreas Hänsler (andreas.haensler@hydrology.uni-freiburg.de)

**Abstract.** Spatially explicit quantification on design storms are essential for flood risk assessment and planning. Due to the limited temporal data availability from weather radar data, design storms are usually estimated on the basis of rainfall records of a few precipitation stations only that have a substantially long time coverage. To achieve a regional picture these station based estimates are spatially interpolated, incorporating a large source of uncertainty due to the typical low station density, in

particular for short event durations.

In this study we present a method to estimate spatially explicit design storms with a return period of up to 100 years on the basis of statistically extended weather radar precipitation estimates based on the ideas of regional frequency analyses and subsequent bias correction. Associated uncertainties are quantified using an ensemble-sampling approach and event-based bootstrapping.

With the resulting dataset, we compile spatially explicit design storms for various return periods and event durations for the federal state of Baden Württemberg, Germany. We compare our findings with two reference datasets based on interpolated station estimates. We find that the transition in the spatial patterns of the design storms from a rather random (short duration events, 15 minute) to a more structured, orographically influenced pattern (long duration events, 24 hours) seems to be much more realistic in the weather radar based product. However, the absolute magnitude of the design storms, although bias-

corrected, is still generally lower in the weather radar product, which should be addressed in future studies in more detail.

## 1 Introduction

In the light of flood risk preparedness preparation and climate change adaptation planning there is a rising need for reliable information on the regional to local impacts of urban and sub-urban storm flows (e.g. European Flood Directive: EC, 2007 or 'Guidelines of heavy rainfall management for the federal state of Baden Württemberg': LUBW, 2016 - in German only). This

information is usually provided based on data from hydrological and hydraulic modelling chains, which themselves need spatially homogenized information on the magnitude of design storms for various duration and frequencies as input data.

In order to be able to provide reliable information, design rainfall estimates have to be based on sufficiently long time-series of rainfall observations from climate stations at a high temporal resolution (e.g. Charras-Garrido and Lezaud, 2013). Especially for the estimates of rare events ($T_r \geq 100a$) this restricts the analyses usually to a rather limited number of precipitation stations,

hence requiring substantial spatial interpolation among the few stations to produce regionalized maps. A further issue when

dealing with long-term station data is the non-stationarity requiring an adaptation of the extreme value analyses (e.g. temporally dependent location parameters of the Generalized Extreme Value distribution, Cheng et al., 2014)

Apart from station data temporally and spatially homogenized and station-adjusted precipitation data from weather radar become increasingly available and have been used in the analysis of design storms (e.g. Overeem et al., 2009; Haberlandt and Berndt, 2016; Panziera et al., 2016; Pöschmann et al., 2021). The main advantage of using weather radar data is the provision of a spatially complete picture of storm events on various temporal and spatial scales, as many short-term and small scale storm events are not captured by the typical network of precipitation gauges (Lengfeld et al., 2020). Hence, design storm estimates based on weather radar data are supposed to provide a more reliable spatial picture than interpolated station data.

One serious drawback of this approach, however, is the lack of long-term weather radar records as spatially and temporally consistent data is only available for the last two decades (e.g. Saltikoff et al., 2019). Although recent studies have shown that statistical techniques are available to estimate design storms (with return periods in the range from 50 to 100 years) on the basis of shorter data series (e.g. Zorzetto et al., 2016), they still have larger uncertainties when compared to estimates on data series equal/longer than the respective return periods.

In order to overcome short records (or ungauged sites), regional frequency analysis is often used for rainfall as well as for discharge records. Based on the so called region of influence (ROI) approach (Burn, 1990), the records of a target station are extended by pooling data from neighbouring stations located within a target-station specific region. While numerous applications of regional frequency analysis are reported for station data (e.g. Gaál and Kyselý, 2009; Requena et al., 2019), fewer examples are available for the extension of time series from weather radar. Goudenhoofdt et al. (2017) based a regional frequency analysis over Belgium on pooled radar data time-series with a sampling scheme considering radar cells in a radius of 10 km around the target cell for the extension of the precipitation records. While in general their approach lead to promising results, the radial sampling scheme lead to some artificial circular pattern in the final product and only defines similar regions based on distance alone.

A slightly different approach to conduct a regional frequency analysis is the spatial bootstrapping method (e.g. Uboldi et al., 2014). For a specific station/cell a large number of samples are established by the repeated sampling of independent events from surrounding stations/cells. This approach was recently applied to 11 years of radar data (spatial resolution of 4 km x 4 km) over the state of Louisiana, US (Eldardiry and Habib, 2020). Also in this study, the cell specific ROI, out of which the samples were pooled, was defined by the distance to the target cell. For each cell they set up 500 samples with a sample size of 11 events (in order to equal the actual number of years), each. They found that the method can provide a robust representation of extreme precipitation which is less affected by single outlier events than a non-regional pixel based approach. However, when compared with station based data, the re-sampled weather radar data has a tendency to underestimate the station records. Reasons for this could be on one hand that the definition of the target cell specific ROI based on the distance only might not be sufficient, but other factors (e.g. elevation, climate) as it is usually done with station data (e.g. Uboldi et al., 2014) should be incorporated as well. Also the fact that each sample only considers 11 events could be a source of uncertainty.

On the other hand, a general 'bias' in the weather radar when compared with stations is visible, generally increasing with
rainfall intensity (e.g. Schleiss et al., 2020, Kreklow et al., 2020) as the radar precipitation is an indirect product (derived from
reflectivity) integrated over a larger area. This fact is another serious drawback when using radar data for the estimation of
design storms. A common approach to correct for such structural biases is the so called bias correction approach (see e.g.
Maraun, 2016 for a review on bias correction) developed in climate impact research, but previously applied to weather radar
data (Rabiei and Haberlandt, 2015). The basic idea behind bias correction is that structural biases in the data are removed while
the specific characteristics (either spatial or temporal) are kept.

We believe that combining regional frequency analysis with bias correction could be a promising approach to generate a robust
radar-based dataset for the spatially explicit estimation of design storm events. In our study, we apply a ROI based approach
to extend a climatological record of 19 years of spatially and temporally homogenized weather radar data in combination with
a station based bias correction. We focus our study regionally on the federal state of Baden Württemberg (BaWu), Germany
as we have two station based, regionally interpolated design storm products available for this region that can be used to evaluate
the newly generated design storm product based on weather radar data. Furthermore, BaWu is topographically quite complex
with an elevation range from 90 m to 1495 m (Fig. 1a), leading to spatially rather inhomogeneous rainfall patterns (see Fig. 1a
and Fig. 1c).

## 2 Data and Methods

### 2.1 Radar-based rainfall estimates

We use the spatially and temporally homogenized climatological precipitation radar product of the German Weather Service
referenced as RADKLIM (Winterrath et al., 2017) that is available as quasi gauge-adjusted five-minutes precipitation product
(RADKLIM_YW_V2017.002; Winterrath et al., 2018). This data consists of post-processed (artefact and attenuation
correction) and station adjusted (but only hourly values) precipitation rates on a 1km x 1km grid for the time period from 2001
to 2019. To be able to directly compare our data product to a station based spatially interpolated data product (see section 2.2.2
below) we only use data for the (summer) months from April to October. Furthermore, the increased uncertainty connected to
the measurement of solid precipitation can be avoided when focussing on the summer season only.

### 2.2 Station based reference data

For an independent reference we use two spatially interpolated design storm estimates based on station data. Both datasets are
frequently used by practitioners in Germany. Both datasets are based on a limited number of stations only and hence, a
substantial spatial interpolation effort was necessary to provide a map of design storms on a regional scale.

### 2.2.1 KOSTRA

The KOSTRA dataset (KOSTRA-DWD-2010R, Junghänel et al., 2017) was compiled by the German Weather Service and can be seen as the national standard with respect to design rainfall in Germany. It provides design rainfall estimates for the whole of Germany for various return periods and event durations. KOSTRA is based on station data for durations below 24h and the raster based REGNIE (Regionalisierte Niederschlagshöhen, DWD,2017; in German only) daily precipitation dataset for longer durations. The temporal record covered by the data products (station and REGNIE) is from 1951 to 2010. Design storms are locally estimated for four different event durations (D=15min, 1h, 12h and 72h) by applying a two parameter GEV distribution to the event data. Design storms for other durations are interpolated from these four durations. In order to map the data to Germany, the local design rainfall estimates are spatially interpolated to a grid on the scale of about 8.2km x 8.2km. It has however to be noted that in the case of the station data used for the durations below 24h only a very limited number of stations (only 56 stations cover the whole period; 94 stations cover the period after 1961) are available for Germany.

### 2.2.2 BW-Stat

Due to the limited spatial resolution of KOSTRA an additional station based dataset (available on 1km x 1km) has been recently compiled for the federal state of Baden Württemberg (subsequently referred to as BW-Stat; Steinbrich et al., 2016 - in German only). This dataset provides the basis of the state's environmental agency for the management of heavy rainfall and resulting pluvial floods in municipalities (LUBW, 2016; in German only). Since the focus is on short to medium range storm events dominated by convective events, only the extended summer season (April to October) was considered for creating the BW-Stat dataset, representing the fact that the extended summer season is the main season for these kind of storm events (e.g. Ruiz-Villanueva et al., 2012; Haacke and Paton, 2021). Nevertheless, the BW-Stat dataset represents design storm estimates for event durations from 5 minutes to 24 hours, since also heavy rain events of rather frontal nature, characterized by longer time durations but still substantial spatial variability can occur in-between the beginning of April and the end of October. Like KOSTRA, this dataset is also based on station-specific local design rainfall estimates which were spatially interpolated using a multi-linear regression approach. The finer resolution of BW-Stat when compared to KOSTRA could be achieved by incorporating data from more stations and other precipitation networks than in KOSTRA into the analyses. The length of the time series, however, varies between 4 and 55 years, with 90% of the stations having 18 or less stations years. Also the temporal coverage differs substantially in-between the stations with some reaching back until the early 1960s but the majority of the stations covering the period after 2000 up to the year 2014. In order to set up a robust data base at each of the locations despite the large heterogeneity in the length of the station records, a ROI based events pooling approach (similar to the one described in this paper – see section 2.3) including neighbouring stations at similar altitudes was used. However, due to the limited station density and the fact that generally stations at similar altitudes are pooled together, the horizontal distance between the pooled stations is generally much larger than in the RADKLIM case. Especially over the mountain regions of the Black Forest, Swabian Jura and Alpine Foothills (see Fig. 1a) the horizontal distance in-between the stations can be up to 80km. It further

has to be mentioned that in the final product all design rainfall values below/above the 5$^{th}$/95$^{th}$ percentile (spatially) have been
set constant (to the 5$^{th}$/95th percentile) by the developers of the dataset in order to prevent for extremely low/high outliers.

In order to estimate design storms the concept of partial series was applied to identify heavy rainfall events and a three parameter Generalized Pareto distribution was applied. For details see section 2.3.3 below, since we use the identical approach in order to make our data set directly comparable to the BW-Stat data. Also BW-Stat design storms are available for different return periods and event durations (5 minutes to 24 hours). To allow a direct comparison with the radar based design storm
estimates, the BW-Stat data was spatially re-interpolated to the radar grid, using the multi-linear regression based interpolation process and station data of the original product.

## 2.3 Data preparation and extreme value analysis

To estimate design storms with a return period of up to 100 years from the available 19 years of RADKLIM data, we developed and applied a multi-step data processing procedure. The data preparation and subsequent extreme value analysis (EVA) was
conducted separately for four different event durations D (15, 60, 360 & 1440 minutes). Unlike KOSTRA, no interpolation has been applied in-between the four event durations. An overview of the complete data processing chain is given in the form of a flow chart depicted in Fig. 2. Below, we describe the data processing in more detail.

### 2.3.1 Calculating event precipitation and selection of independent events

Starting with the original five-minutes gridded RADKLIM data, we first calculate cell specific precipitation event sums P$_{SUM}$
for each of the four durations D using the method of running sums.

$$P_{SUM}(\text{t}) = \sum_{i=0}^{\frac{D}{\Delta t}} P(t + i) \text{ ; with } \Delta t = 5\text{min} \tag{Eq 1}$$

From this dataset we then select the 350 largest and temporal independent precipitation events. The number of 350 events has been chosen to guarantee that the sample size is large enough for the subsequent EVA, already knowing that not all events will be included in the EVA. Temporal independence of the individual events is ensured by selecting only events that are at least
48 hours apart. This time spacing is applied for all durations, although for short duration events this might be a rather conservative definition of independence. For the selection of the events we rank the precipitation events from the largest to the lowest events and select the rank 1 event from the full event data set (see EQ 2). Subsequent to this, we remove all events from the data set that are within a 48hour range of the rank 1 event and select again the rank 1 event from the remaining event data set. This procedure is repeated until we have identified 350 events for each of the radar cells and the four durations.

$$P_{MAX}(X_i, t) = \begin{bmatrix} \max(P_{SUM}(t)), X_i \\ \vdots \\ \max\left(P_{SUM}(t)\right), X_{i+1} \end{bmatrix} \text{; with } \Delta t_{xi; \, x(i+1)} \geq 48\text{h} \tag{Eq 2}$$

### 2.3.2 Regional sub-sampling

We assume a storm event with a return period of 100 years to represent the upper end of our analysis. Therefore, we aim for a target length of the underlying time-series of about 100 years of rainfall data to meet the requirements for a profound EVA,

although we are aware of the fact that a 100yr event is not necessarily present when analysing 100 years of data. Given the 19 years of RADKLIM data, we need to pool for each radar cell (cell of interest, COI) the data from four additional radar cells to statistically extent the RADKLIM data series to a respective length (95 years).

Based on the ROI concept we defined for each COI a specific sampling area with a specific sampling probability for each cell assigned. The definition of the COI specific sampling area has to fulfil two criteria. On the one hand, the specific sampling

area has to be located in close proximity (in terms of horizontal as well as vertical distance) to the COI in order to be spatially representative. On the other hand, we also want to make sure that we sample additional rainfall events or intensities not present in the COI, so we have also to make sure that the sampling happens not too close to the COI.

For each COI we first estimated a specific sampling area based on the radial and vertical distance of an individual radar cell to the COI. The underlying spatial sampling probabilities $S_{Prob}$ are separately assigned for the radial (circ) and altitudinal (oro)

sampling each following a normal distribution $N(\mu,\sigma)$ and normalized to its respective maximum.

$$S_{Prob}(x)_{circ;\,oro} = \frac{f(x)_{circ;\,oro}}{\max(f(x)_{circ;\,oro})}\,;\ \text{ with } f(x)_{circ;\,oro} = N(\mu_{circ;\,oro}, \sigma_{circ;oro}) \qquad \text{(Eq 3)}$$

The respective parameters underlying the sampling probabilities are summarized in Table 1. With respect to the radial distance we set the maximum sampling radius ($R_{Max}$) to 25 km to somehow reflect the typical area impacted by a convective cell in Germany (~25 to 40 km for hourly events in the summer season in BaWu, Lengfeld et al., 2019) but still keep the spatial

representation of the sampling region for the COI.

In a subsequent step, we combine the radial and vertical based sampling probabilities into a COI specific, normalized final sampling probability $S_{Prob}$.

$$S_{Prob}(x) = \frac{f(x)}{\max(f(x))}\,;\ \text{ with } f(x) = S_{Prob}(x)_{circ} + S_{Prob}(x)_{oro} \qquad \text{(Eq 4)}$$

For each COI we now randomly sample four additional cells out of all cells with $S_{Prob} > 0.8$ ($P_{Tresh}$). This sampling is conducted

iteratively and each time after a sample is drawn, $S_{Prob}$ of all cells in a radius of 4 km to the sampled cell is reduced to a value below $P_{Tresh}$. This is done in order to prevent that neighbouring cells are sampled since this would limit the number of additional rainfall events. A graphical illustration of the sampling process for one specific COI is given in Fig. 1 (panels bI to bIV).

After finishing the sampling process for a specific COI, we merge the data of the 350 independent events of the 5 cells (COI plus the additional four sampled cells) into an extended set of 5x 350 events. Since after the merging the temporal independence

of the events is no longer guaranteed, we repeat the event selection procedure described in section 2.3.1. The resulting dataset is then used as input data for the subsequent EVA and bias correction.

Since we allow random sampling out of all cells with $S_{Prob} > P_{Tresh}$, repeating the sampling process will likely result in a different set of sampled cells for a given COI. Hence, it is possible to follow an ensemble approach for the sampling to be able to quantify the sampling uncertainty. Following this, we repeat the sampling process for each COI ten times. However, to minimize the effect of duplicated samples (cells) in the individual ensemble members at a given COI and therefore maximize the effective ensemble size, only the five ensemble members with the lowest number of cell duplicates at each COI are selected.

$$Ens_{x_i} = min(Ens_{x_i} \in [Ens_{x_i}, ..., Ens_{x_j}]) \qquad \text{(Eq 5)}$$

It has to be noted that the regional sub-sampling is not adapted for the four event durations. However, this does not imply that the identical events are analysed since the sampled cells can contribute different numbers of events for each of the event durations, depending on the actual rain amounts. The main reason behind keeping the sampling process constant is that we wanted to make sure that any change in the spatial patterns of the design storms between the different event durations is not affected by the sampling process, but by the rainfall data itself.

In order to support the choice of the underlying sampling parameters $\mu$, $\sigma$, $R_{Max}$ & $P_{Tresh}$, we analysed the relative contribution of each sampled cell to the final data set, the distance of the sampled cells to the COI as well as the effective ensemble size (see Fig A1, upper panels). We find that for most parts of BaWu the effective ensemble size is five. Also the frequency of occurrence as well as the distance of the sampled cells to the COI are in close proximity to what is theoretically expected.

### 2.3.3 Extreme value analysis

We follow the same approach as applied in Steinbrich et al. (2016) to directly compare our data product with the BW-Stat dataset and it follows the guidelines for EVA given by the German Association for Water, Wastewater and Waste (DWA, 2012). As input data we use the set of 350 precipitation events for each duration generated through the regional sub-sampling process. For each radar cell these events reflect the maximum independent events of a data series of 95 (5 x 19) artificial years. Each radar cell and each of the five ensemble members is hereby treated as an individual station. Although a time series of 95 years was generated, it has to be kept in mind, that the events are selected based on 19 years of weather radar rainfall estimates, only. Hence, the concept of partial series (value over threshold concept) instead of annual series is applied to select the events for the EVA. The threshold value varies from cell to cell and is estimated to be the value that has a return period of 1 year using the approach of plotting positions $T_k$ for each element k of the partial series (with k =1 representing the maximum event for the specific cell, duration and ensemble member within the 95 artificial years).

$$T_k = \left( {L + 0.2}/{k - 0.4} \right) * (M/L) \qquad \text{(Eq 6)}$$

with M as the length of the time series in years (95 years in our case). L is the total number of independent events finally included into the EVA which is in our case estimated by e (Euler's number) times the number of years equals 258 events.

For all 258 events with rainfall rates equal to or above the threshold value, the Generalized Pareto distribution (GPD; Eq. 7 - see also e.g. de Zea Bermudez and Kotz, 2010 for details on the parameters of the GPD) is fitted in order to be able to calculate precipitation rates for various return periods. The three (location, scale and shape) parameters describing the GPD are estimated using the L-Moment parameter estimation method. Note that the estimation of the GPD parameters is done individually for each event duration, radar cell and ensemble member. Also the application of the GPD and the fitting process is similar to the approach used for the generation of the BW-Stat dataset and enables the direct comparison between our dataset and the BW-Stat estimates.

$$F_{(\mu,\sigma,\varepsilon)}(x) = 1 - \left(1 + \frac{\varepsilon(x-\mu)}{\sigma}\right)^{-1/\varepsilon} \quad \text{with } \mu \text{ as location, } \sigma \text{ as scale and } \varepsilon \text{ as shape parameter} \qquad \text{(Eq 7)}$$

### 2.3.4 Bias-correction of RADKLIM Data

As mentioned in the introduction rainfall estimates from weather radar are known to frequently underestimate the magnitude of extreme rainfall events when compared to station data. This is usually caused by the fact that radar measurements represent an integrated measurement of 1km x 1km while station data is a point measurement, but also other effects like an underestimation of high-intensity rainfall estimates using fixed Z-R relations for typical convective and stratiform events may play a role (e.g. Thorndahl et al., 2014). In order to compensate for such structural biases, we decided to match the magnitude of 1yr design storms of the BW-Stat dataset and the radar data. The decision to base the correction on the location parameter (which can be taken as a proxy for a 1yr event) is motivated by the fact that also the time series of the stations underlying the BW-Stat dataset are rather short themselves (see section 2.2.2.). While the location parameter can still be derived in a rather robust manner in both datasets the scale and especially the shape parameters would be more affected by the regional sub-sampling applied.

To achieve this match of the location parameter of the two datasets, a quantile mapping approach (e.g. Cannon et al., 2015) was applied. The basic principle behind quantile mapping is that the cumulative frequency distribution functions (CFDs) of the two datasets are equalled via a transfer function.

$$\hat{x}_{rad} = F_{stats}^{-1}\{F_{rad}[x_{rad}]\} \qquad \text{(Eq 8)}$$

The major advantage of the QM approach is that it corrects the bias for the whole CFD but keeps the respective spatial pattern of the data. For each station within the analysis region we select the location parameter of the closest four radar cells. The respective spatial CFDs are calculated for all stations and their corresponding cells for each duration and ensemble member separately. The transfer function between the two CFDs is estimated on the basis of 100 discrete bins and is then applied to the CFD of the location parameter of the full radar data set (again separately for each duration and ensemble member).

### 2.3.5 Calculation of design storms and uncertainty estimate

All radar based design storms are calculated based on the corrected location parameter, however the shape and scale parameters of the GPD have not been corrected in order to keep the consistency within the data. The design storm estimates form bias-corrected weather radar based GPD parameters is referred to as RAD-BC whereas the non-bias-corrected version is named RAD.

In order to estimate the uncertainty of the estimated design storms of RAD-BC we apply a twofold uncertainty estimation. First we quantify the uncertainty related to the spatial sub-sampling via the application of an ensemble approach caused by the five-member ensemble generated in the sampling process. Second, we can estimate the uncertainty of the EVA parameter fitting. This is done by applying a classical bootstrapping method for each duration, cell and ensemble member to generate 1000 random samples of the events identified for the extreme value statistics. This results in a final total ensemble of 5000 parameter estimates for each cell and duration, hence allowing to explicitly assign confidence intervals to the estimated design storms. The advantage of the chosen approach is that it allows to eventually separate between the uncertainty range resulting from the spatial pooling and the parameter fitting. While the latter is represented by the full range of all 5000 members, the uncertainty related to the pooling can be estimated by the span within the five ensemble members.

## 3. Results

### 3.1 Bias correction

The impact of the quantile based correction of the location parameter is depicted in the form of spatial CFDs in Fig. 3. While the uncorrected radar data substantially underestimates the 1yr design storms, the bias corrected version mimics (by design) almost perfectly to the station data when only the grid cells representing station points are included (upper row). Considering all of BaWu the comparison between interpolated station data and bias corrected radar data leads to slightly larger differences (bottom row) also partly resulting from the assumptions behind the spatial interpolation of the station data. It has to be noted that both, BW-Stat and RAD-BC estimates, still show substantially lower rain rates for the 1yr design storms than the KOSTRA reference dataset, for most parts of the distribution. The overestimation of extremes in the case of shorter event durations can be attributed to the lower spatial resolution of KOSTRA. Linked to this is also the substantially lower variability of KOSTRA, when compared with the other two datasets.

What should be kept in mind is the fact, that the applied bias correction is not having the same effect for longer return periods. Correcting 1yr design storms only means that a certain rain amount is added to all events included in the EVA, hence, the relative contribution of the bias correction decreases for less-frequent design storms (see the differences between RAD and RAD-BC in Fig. 5).

### 3.2 Comparison of design storms

The spatial patterns of a 100yr design storm for four different selected event durations (15, 60, 360 and 1440 minutes) for the two station based reference datasets (KOSTRA, BW-Stat) as well as for the bias-corrected and re-sampled RADKLIM dataset (RAD-BC) are depicted in Fig. 4. Additionally, the absolute difference between BW-Stat and RAD-BC datasets is depicted. Note that the RAD-BC dataset represents the ensemble mean of the five individual sample products and that the data is spatially smoothed with a 3 by 3 cell filter to avoid single outliers. For comparison we compiled the identical figure for a 1yr design storm (see Fig. A2 in the appendix).

In the KOSTRA dataset orographic induced patterns with elevated storm intensities along the Black Forest mountains and the Swabian Jura as well as the Alpine foothills (see Fig. 1a for regional specification) in the far south east can be seen for short and long duration events. This rather stabile pattern can be expected since the z-coordinate was incorporated in the interpolation of the station data (Junghänel et al., 2017). Further, the 360-minute design storm in KOSTRA is interpolated from the 60 min and 12h (not shown) design storms and also the 24h design storm represents an interpolated value (interpolation between 12h and 72h design storms). In BW-Stat, the Black Forest region is also characterized by high-intensity design storms for both, short and long duration events. However, especially for events with longer duration BW-Stat shows very dominant, high-intensity design storms in a region located between the Lake of Constance and the Black Forest, usually known to represent rather a rain shadow area due to fronts moving in from the west (see Fig. 1c).

The spatial patterns in the RAD-BC dataset differ quite substantially from the patterns of the two station based reference datasets and also shows a distinct pattern change between short and long-duration events. While the spatial patterns of the 15 and 60 minute 100yr design storms show no relation to the orography or orographically induced rainfall patterns (but a slight north-south gradient) it changes in the case of the 1440 minute 100yr design storm events to a picture very similar to the April to October mean rainfall distribution. This finding is supported by a cross correlation analyses between the RAD-BC data and the mean rainfall estimates from REGNIE which reveals an increase in the correlation coefficient from r=0.25 (15 minute events) to r=0.75 (1440 minute events). In the case of BC-Stat r remains below 0.6. The spatial pattern of RAD-BC design storms is much more in line with what is expected from the underlying processes representing pure convection triggered, small scale feature for short duration event and more organized larger scale frontal systems for longer duration events (Lengfeld et al., 2019; Kaiser et al., 2021). Interestingly, the spatial pattern in the BW-Stat dataset is following this behaviour in the case of a 1yr design storm (similar to RAD-BC, see Fig. A2). This can be attributed to the fact that the 1yr design storm is less affected by the spatial pooling than the 20 or even 100yr design storms. Since the spatial pooling in the BW-Stat dataset is based on a limited amount of stations the underlying sampling area can be rather large. In combination with the spatial interpolation this leads to the effect that for low frequency design storms, large areas of BaWu are influenced by events of single stations.

With respect to the absolute values, the direct comparison of BW-Stat and RAD-BC design storm intensities reveal that there are regions with substantially larger intensities in the RAD-BC dataset (e.g. especially in the far south east for the 1440 minute

events) due to the difference in the spatial patterns. Also in case of the 1yr design storms (Fig. A2) RAD-BC shows generally

larger intensities than present in BW-Stat over the mountainous regions, although this is most probably largely affected by the

fact that in BW-Stat all values above the 95[th] percentile were set to the respective percentile value.

However, when integrated over the whole study region RAD-BC shows lower rainfall magnitudes for 20yr and 100yr design

storms than the two station-based reference datasets. In Fig. 5 we depict the spatial CFD of the different datasets for the

different durations and two (20yr and 100yr) return periods. To illustrate the effect of the bias correction, the non-bias-corrected

radar dataset (RAD; green line) is also shown. Additionally, the respective confidence interval for the RAD-BC dataset (see

section 3.3 below) is included.

Apart from the very high and low percentiles, the ensemble mean of the RAD-BC storm events is about 5 to 15mm lower than

the respective rain rate of BW-Stat. Nevertheless, the uncertainty range spanned within the two station based reference datasets

is quite large itself. While there are cases where the KOSTRA dataset lies within the confidence interval of the RAD-BC

dataset (e.g. 100yr design storm with duration of 15 min), the difference to KOSTRA is sometimes even larger than to BW-

Stat (e.g. 20yr design storm with duration of 360 min). For the BW-Stat data we additionally can estimate the error (RMSE)

resulting from the spatial interpolation using a cross-validation approach directly at the location of the stations (light-red band).

Although the RAD-BC dataset is mostly at the lower end of the uncertainty range from the spatial interpolation, it becomes

obvious that the uncertainty from the spatial interpolation of BW-Stat is in important factor that can be circumvented when

using a spatial rainfall product.

## 3.3 Uncertainty of design storms

In order to be able to quantify the uncertainties for the newly developed RAD-BC dataset we conducted a twofold uncertainty

analysis based on an ensemble based cell-sampling approach and classical bootstrapping for the identification of parameter

uncertainty. The confidence interval in Fig. 5 is defined by the 5[th] and 95[th] percentile of the large data sample generated by

1000 bootstraps runs for each of the 5 ensemble members representing a combination of both sources of uncertainty. The

confidence band of the CFD spans about 5mm in the case of 20yr design storms and about 10mm in the 100yr case. The range

of the five ensemble members only (without bootstrapping) is defined by the stippled line and accounts for a large amount of

the total uncertainty band. This demonstrates the importance of the ensemble based sampling approach.

The spatial patterns of the 5[th] and 95[th] percentile are rather similar to the patterns of the ensemble mean (see Fig. 6), and the

uncertainty range of the respective rain rate is for most regions between 15 and 20% in the case of 60 minute events and

between 10 to 15% in the case of 1440 minute events, with relatively larger ranges in regions with lower values for the mean

storm intensity. However, there are certain spots (e.g. the northern parts of the Black Forest in the case of a 100yr 1440minute

design storm - framed with a dashed square in Fig. 6 - or various smaller regions in both examples) that have a slightly larger

uncertainty range, although the mean storm intensities are large as well. In order to reveal the uncertainty resulting from the

ensemble sampling we highlighted regions with a relatively large (> 65% of the range) ensemble spread. Generally, the

contribution of the sampling uncertainty is larger in regions with a lower overall uncertainty range. However, there are various

spots that are dominated by the sampling uncertainty that have a relatively larger overall uncertainty range. An example for this is the previously mentioned enhanced uncertainty in the northern Black Forest region that seems a to be substantially influenced by sampling uncertainty in its eastern parts. This can be seen as an indication for a rather inhomogeneous pool of heavy rainfall events sampled in this region.

On top of these directly quantifiable uncertainties there is also the uncertainty related to the choice of the sampling parameters underlying the regional pooling. In the lower part of Fig. A1 we compare the mean rainfall sum of the maximum 10 events (R10Max) of the original (RAD) as well as the spatially resampled but not bias corrected (RAD_resampled) radar dataset to the data of a multi-parameter ensemble that has been generated by systematically varying the sampling parameters (see right part of Table 1 for the parameter range). While the sampling parameters underlying RAD-BC maintain the balance between adding new events but still reflect the spatial distribution of RAD, increasing the potential sampling area (e.g. via lowering $P_{Tresh}$ or increasing $\mu$, $\sigma$, or $R_{Max}$) substantially increases R10Max but the spatial patterns start to blur. Selecting the parameters in a way that the potential sampling area is rather small, the spatial patterns are closer to RAD, however, the increase in R10Max is smaller. Additionally, in this case the effective ensemble size is reduced (not shown), since the number of duplicated cells per COI in the different ensemble members is higher.

## 4. Discussion

One of the difficulties of our study is that there is no classical validation dataset available. Although we include two station based gridded design storm products in our analysis, they differ themselves quite largely in both, absolute amounts and spatial patterns. Given the methodological differences of the two datasets, with different number and time coverage of stations, different extreme value statistics and different spatial interpolation methods being the three most important features, these substantial differences between the two reference dataset are not surprising. Especially the different time periods covered by the station data can be a serious source of uncertainty, given the high temporal variability in the occurrence of heavy rainfall events. A recent study based on RADKLIM revealed that the year 2018 was characterized as a year with an exceptional number of heavy rainfall events in Germany (Lengfeld et al., 2021), but no general trend in extreme rainfall events could be identified on the basis of the radar period since 2001. These events from the year 2018, however, are only included in the RAD-BC dataset but not in BW-Stat or KOSTRA.

The major added value of the RAD-BC dataset is the possibility to derive spatially homogenized heavy rainfall estimates for events with a return period of up to a 100 years. A comparison with the station based spatially interpolated reference products revealed that the spatial patterns of the design storms for the four different durations fit much better to the theoretically expected spatial patterns than in the interpolated station products. In KOSTRA, e.g. the stability of the spatial patterns of design storms of different event durations can partly be contributed to the interpolation of values in-between different event durations (see 2.2.1). While this might be beneficial from an engineering perspective, we explicitly calculated the design storms separately

for all durations in order to preserve the spatial patterns of the underlying radar product. In BW-Stat, the subsampling of
stations seems to have a substantial impact on the sequence of spatial patterns for the different durations (see section 3.2)

Although the spatial patterns are identified to be more reliable in RAD-BC, the general tendency to underestimate the magnitude of design storms is something which should be examined in further detail. Given the methodological differences in the datasets a direct one to one comparison is only possible (with certain limitations) with the BW-Stat data. Comparing the non-bias-corrected scale and shape parameters of the GPD fitted to BW-Stat and to an arbitrary ensemble member of RAD-
BC over all of BaWu (Fig. 7, left panels) reveals that for the short durations (15 and 60 min) the scale parameter is lower in the RAD-BC data. For the long (1440 min) events, however, the deviations in the magnitude of the design storms seem to result mainly from the shape parameter which is lower in RAD-BC. This finding again can partly be attributed to the large contribution of single stations to the most extreme events of BW-Stat as a consequence of the subsampling over relatively large regions. This underestimation of the scale/shape parameters in RAD-BC for short/long durations is confirmed when
looking at various topographic sub-regions (Fig. 7, other panels) and other ensemble members (not shown) of the RAD-BC dataset.

The lower values for the scale/shape parameters of RAD-BC can partly also be attributed to the fact that for high rainfall intensities radar data is known to underestimate rainfall amounts due to the fixed Z-R relationship not reflecting changes in rain drop characteristics with increasing rainfall intensities (e.g. Schleiss et al., 2020). A recent comparison of the RADKLIM
data to station data further revealed that fewer heavy rainfall events are detected in RADKLIM than in the station data. The average rainfall amount of a heavy rainfall day (> 20 mm of rainfall) is, however, almost identical (Kreklow et al.,2020).

The general underestimation of heavy rain events in RADKLIM is only partly corrected for by the applied bias correction of the location parameter since it is an additive correction which corrects more frequent events relatively stronger than the less frequent events. Another approach that not only impacts the location but also the scale and shape parameters of the GPD is, to
apply the bias correction in a multiplicative manner. A grid-specific multiplication factor can be estimated on the basis of the uncorrected and corrected 1yr design storms. Applying the resulting multiplication factor to the data would lead to a substantial increase in the rainfall amounts also for the less frequent events (see Fig. A3) but the spatial patterns of the RADKLIM data would be preserved. Nevertheless, one has to keep in mind that a multiplicative correction is disrupting the homogeneity of the sampled events of the radar data, adding much higher rainfall amounts to the more intense rainfall events than to the less
extreme events. It is also questionable if a correction factor derived from the correction of 1yr events can be applied to events with a much lower frequency. Further it has to be kept in mind that the BW-Stat data itself is an indirect product with events pooled from surrounding stations with similar altitude but sometimes rather large distances. In combination with the substantial uncertainty through the interpolation process, this itself could lead to a biased picture in the magnitude of the derived design storms in BW-Stat.
While the location parameter still can be seen as rather robust, it is highly questionable if the derived scale and shape parameters of BW-Stat could be used with the same reliability for the bias correction of the radar data. Fig.7 further reveals that including only the scale parameter as additional parameter into the bias correction might improve the representation of short duration

events in RAD-BC but will not reduce the remaining bias for the 1440 minute events. Looking at the spatial CFD of the BW-Stat scale parameter at higher elevations (upper right panel in Fig. 7), however, reveals a rather inhomogeneous CFD regime most pronounced for the long duration events. This again indicates the weaknesses of the regional subsampling in BW-Stat due to the low station density. Using this as the reference baseline for the QM would impose a large regional heterogeneity to the radar data.

A promising way to proceed without the limitations from the regional subsampling of the BW-Stat data could be to only use a small subset of stations that have a reasonable long record. Based on this subset of data a frequency and duration specific correction function could be developed, which could then be regionally applied to the radar data. However, for BaWu there are only two stations with high temporal precipitation records available with a data series length of more than 50 years (Steinbrich et al., 2016) posing a major challenge for this approach. Another possible approach could be to base the correction on the underlying observed rainfall events itself instead of correcting the parameters of the GPD. This would have the benefit that the high-intensity events would be directly corrected and not derived based on the correction factors estimated for less-intense events. However, the different time periods covered by the station and radar data limit the number of stations and events that could be included in such an analyses.

On top of applying bias correction methods, using a weather radar product that is compiled at a higher spatial resolution and additionally uses an adapted calibration procedure that does not necessarily distort the radar signal to match the station record (e.g. Weiler et al., 2019) could also be a promising approach. While a higher spatial resolution is expected to enhance the measured rainfall amounts due to the lower integration areas, the adapted calibration procedure has the positive aspect that high rainfall intensities captured by the radar are not reduced by nearby stations that are might not affected by the heavy rainfall event itself. However, a real benefit would only be achieved if the deviations between rainfall estimates of weather radar and station data are not increasing with rainfall intensities, which could be reached by a non-static application of the Z-R relation in the weather radar product.

## 5 Conclusions

We present an ROI based approach to prolongate a 19yr climatological weather radar dataset of rainfall estimates in order to enhance its usability for the development of region specific design storm events. The established method has various positive aspects. The main improvement is the development of a spatially homogeneous dataset that allows for the calculation of extreme events without spatial interpolation, which often is the main error source when building a regional dataset based on station data. Moreover, the chosen sampling approach allows to control the sampling region based on physical aspects while preventing artificial circular structures previously reported in literature (e.g. Goudenhoofdt et al., 2017). By the combination of an ensemble-based sampling approach and a bootstrapping based parameter estimation an explicit designation of associated uncertainty ranges is possible, representing a major added value for the application by practitioners.

Nevertheless, the current version of the RAD-BC data preparation method still has some shortcomings that need to be
addressed in the future. While the applied bias correction approach substantially improved the outcome and can be classified
as a robust method, the persisting deviation to the two station based reference datasets is still something that has to be clarified
in the near future. To improve the compatibility with the KOSTRA dataset it might be worthwhile to apply the KOSTRA EVA
to the resampled event database which underlies RAD-BC. Furthermore, the previously proposed training of the RAD-BC
dataset on some high-quality long-term temporally highly resolved station data could be a way forward to further enhance the
credibility of the RAD-BC dataset.

**Author contributions**

AH and MW jointly designed the experiment. All data analyses have been conducted by AH. The interpretation of the results
as well as the drafting of the manuscript was conducted jointly by AH and MW.

**Acknowledgements**

This work was conducted within the research activities on heavy rainfall at the Chair of Hydrology, University of Freiburg,
Funding for these research activities are provided by the State Office for the Environment, Measurements and Nature
Conservation of the Federal State of Baden-Württemberg (LUBW) as well as the Regierungspräsidium (governing council)
Stuttgart.

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

# Figures & Tables

**Table 1: Parameters used for defining the COI specific sampling probabilities as well as the respective parameter range used to estimate the uncertainty related to the sampling parameters.**

|  | RAD-BC subsampling | | | Sensitivity tests (parameter range) | | |
|---|---|---|---|---|---|---|
|  | μ | σ | Maximum radius | μ | σ | Maximum radius |
| **Circle (circ)** | 9 km | 6 km | 25 km | 6 − 12 km | 3 − 9 km | 15 − 35 km |
| **Altitude (oro)** | Altitude of COI* | 50 m | - | Altitude of COI* | 35 − 65 m | - |
| **Prob. threshold** | 0.8 | | | 0.65 − 0.95 | | |

\* Note: For all COI with an altitude above 1150m (~70 cells), μ was set to 1150m instead of the COI altitude to enhance the number of cells available for sampling.

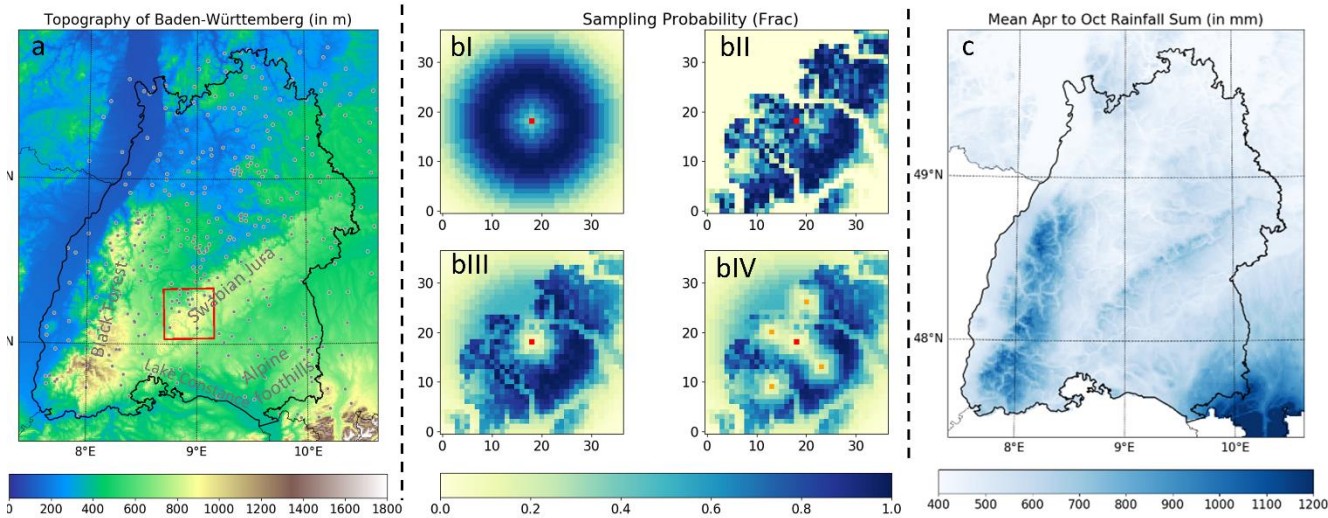

**Figure 1: (a): Topography of Baden Württemberg (BaWu) as well as location of the precipitation gauges used in the BW-Stat dataset and some of the geographical regions referred to in the text. (b): Probability for a specific radar cell to be sampled based on the distance to cell of interest(bI), orography (bII) and orography and distance combined (bIII). Final sampled cells (orange) and reduced probabilities around the selected cells are depicted in panel bIV. All panels reflect the area indicated with a red square in the left part of the figure. The respective cell of interest is marked in red. (c): April to October rainfall sum (1991 to 2020) of the REGNIE (Regionalisierte Niederschlagshöhen) dataset compiled by the German Weather Service.**

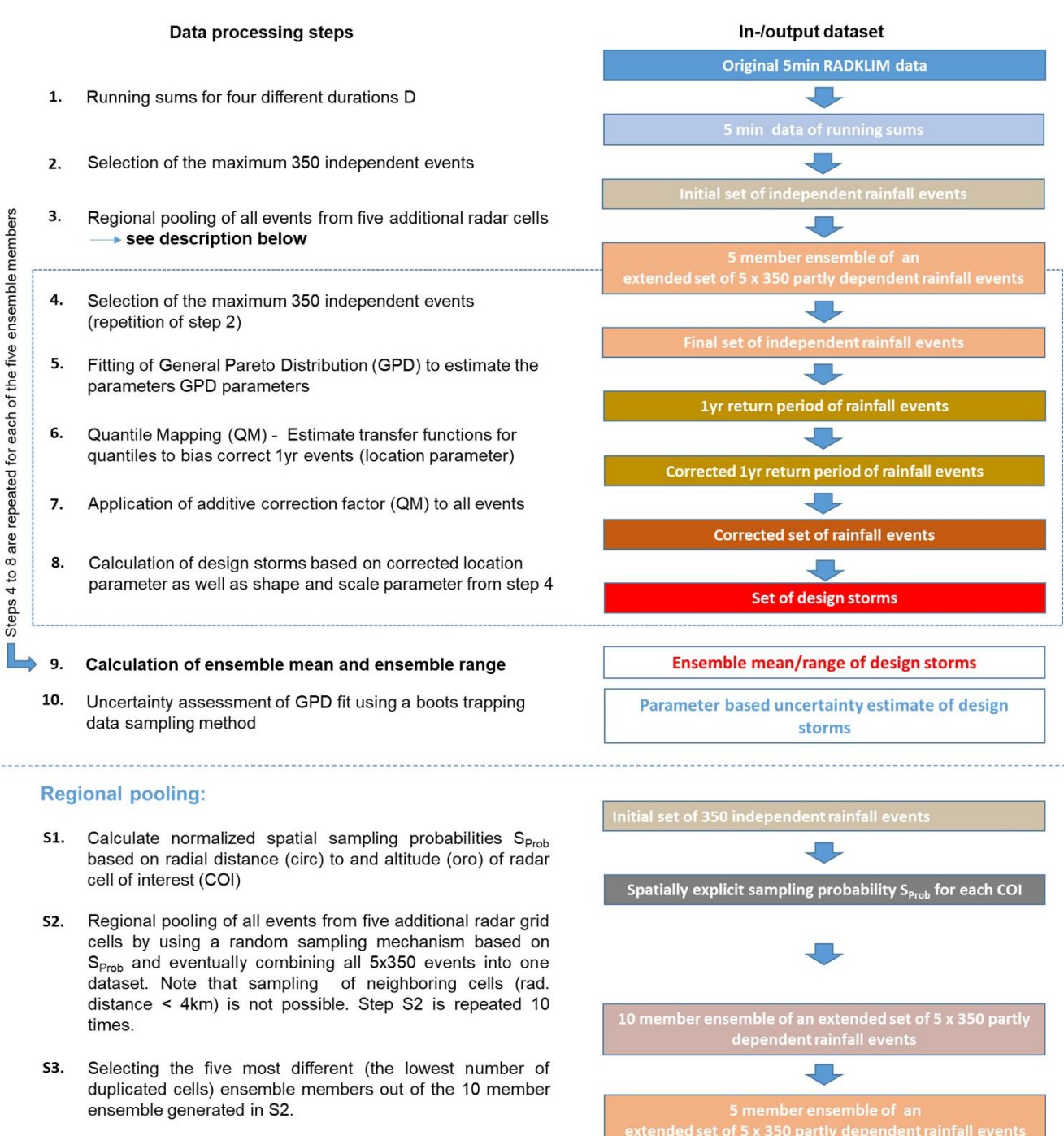

**Figure 2: Flow chart visualizing the data processing chain to establish the RAD-BC dataset. The boxes describe the respective input/output dataset of each data processing steps. Note that the full data processing chain was repeated for each of the four event durations (D=15,60,360 & 1440 minutes) considered in this study.**

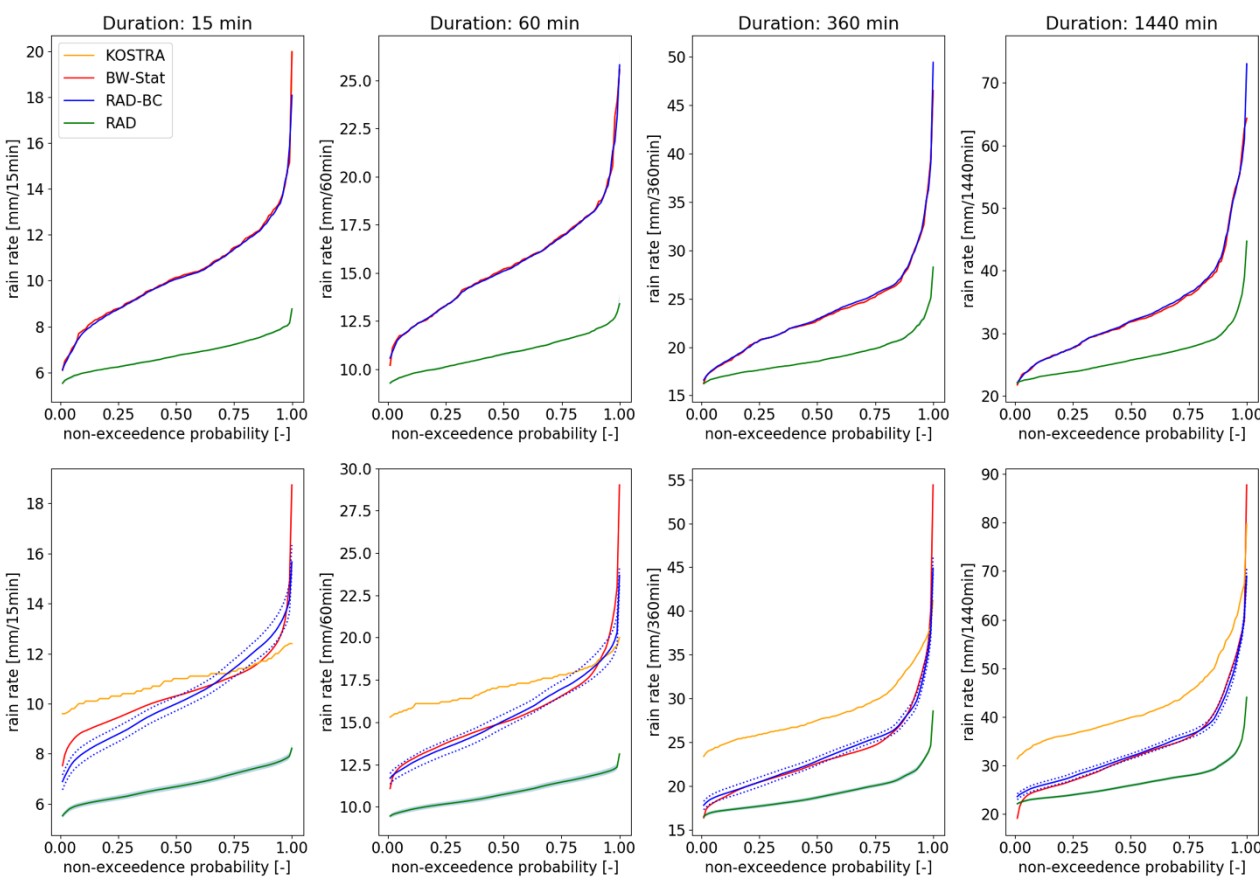


**Figure 3: Spatial cumulative frequency distributions (CFD) of the location parameter for four different event durations when comparing stations and radar data at the location of stations only (upper row) and integrated over the whole of BaWu (bottom row). The dotted blue lines in the bottom row represent the range of the five ensemble members (sampling uncertainty only, no bootstrapping).**


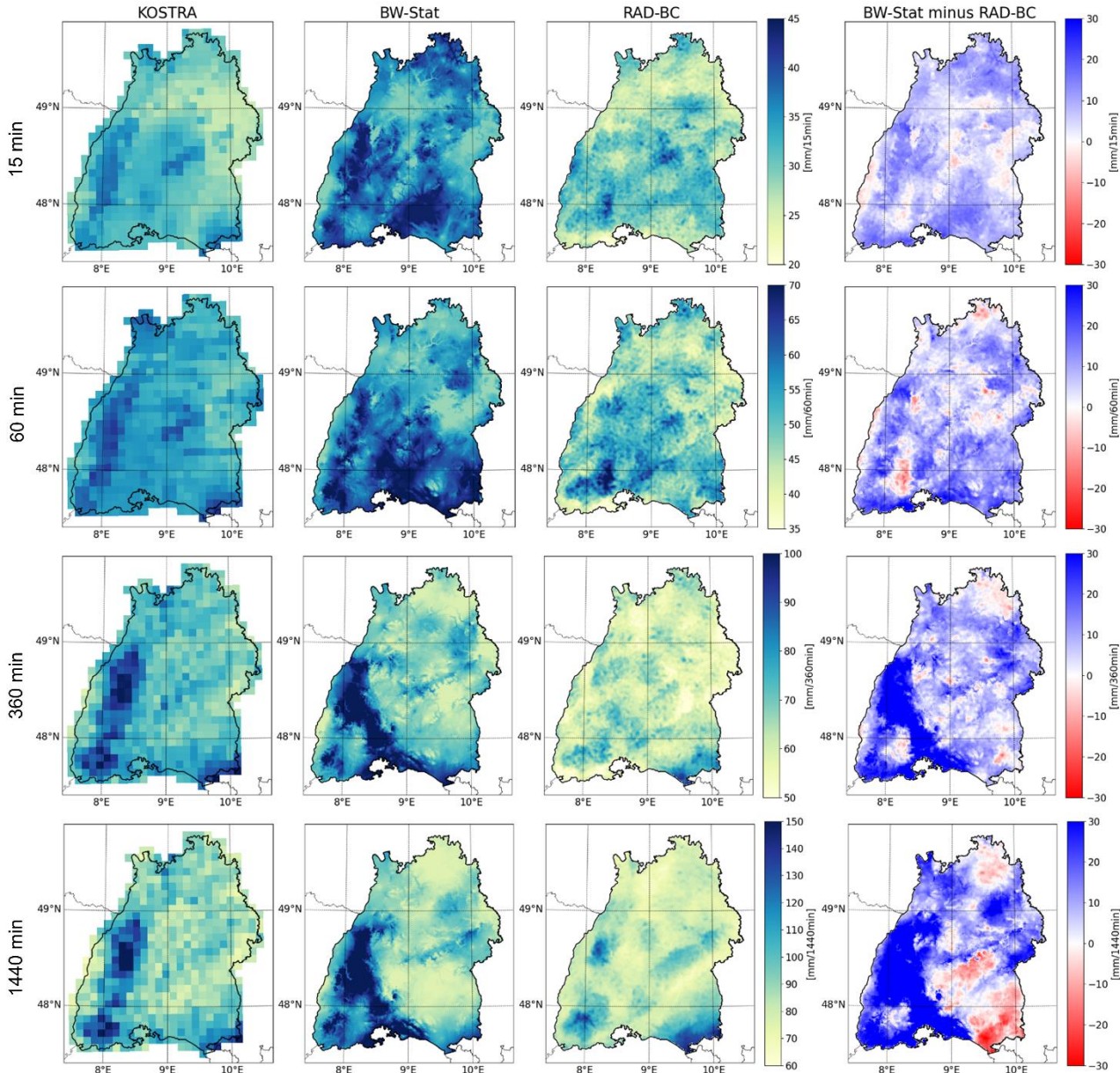

**Figure 4: Magnitude of design storms with a return rate of 100 years for four different event durations (15, 60, 360 and 1440 minutes, depicted in rows) and three different datasets (KOSTRA, BW-Stat, RAD-BC, depicted in columns). Additionally, the difference between BW-Stat and RAD-BC is depicted (right column).**


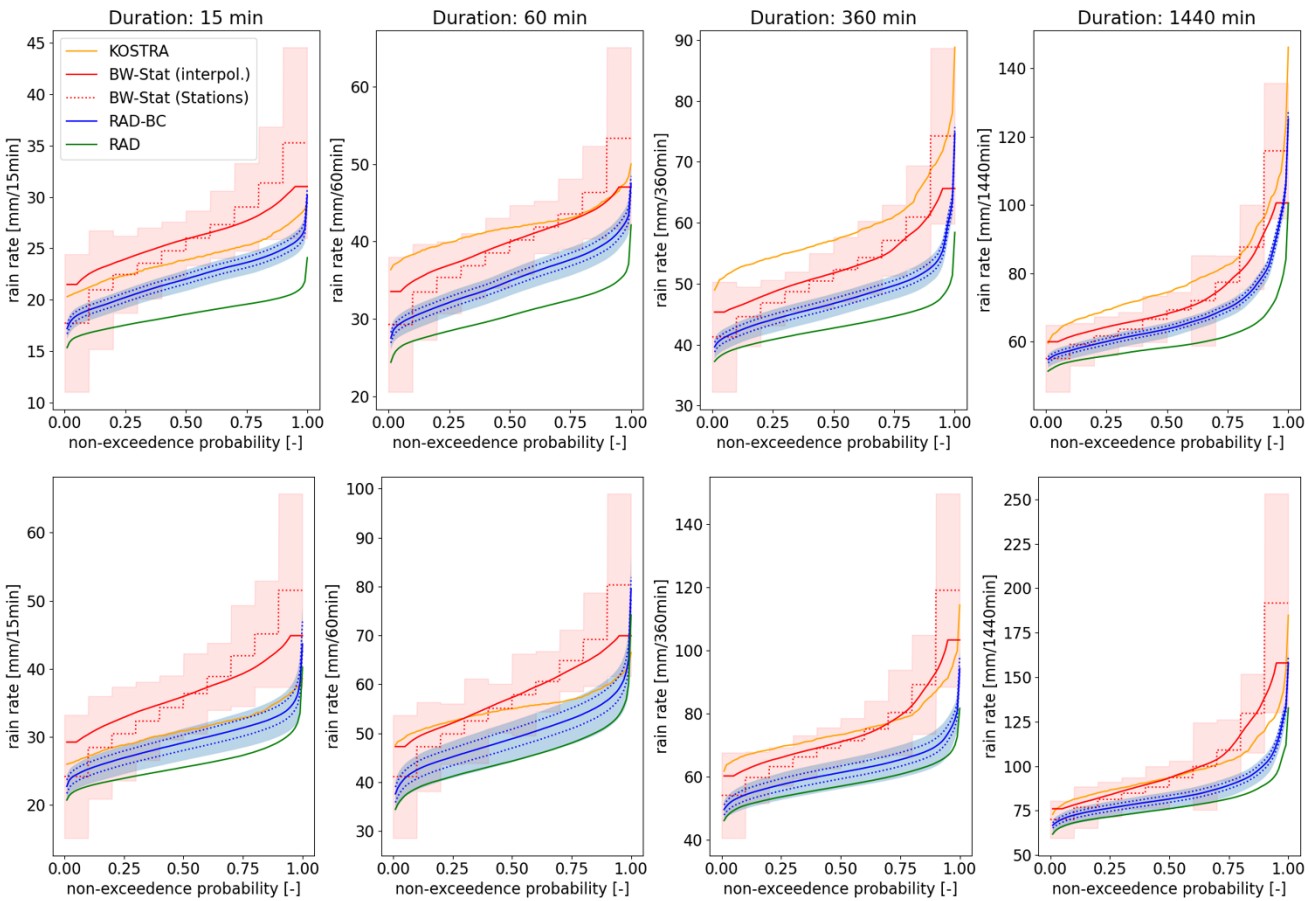

**Figure 5: Spatial cumulative frequency distributions (CFD) of the magnitude of 20yr (upper row) and 100yr (bottom row) design storms for four different event durations and different datasets. The blue shaded range depicts the ensemble uncertainty (5th and 95th percentile of the range from the 1000 bootstraps for each of the 5 ensemble members). The dotted blue lines in the bottom row represent the range of the five ensemble members (sampling uncertainty, no bootstrapping) only. For comparison we added the interpolation error (RMSE) – red shaded area of the underlying stations (red dotted line) of the BW-Stat dataset. Note that for the RAD, RAD-BC, BW-Stat (interpolated) and KOSTRA dataset the CFD are calculated on the gridded data (with fewer grid boxes in KOSTRA) while for the BW-Stat (Stations) data the stations have been binned in ten bins. In the latter dataset all stations are included in the CFD while in the gridded BW-Stat no values above/below the 5th/95th percentile are available.**

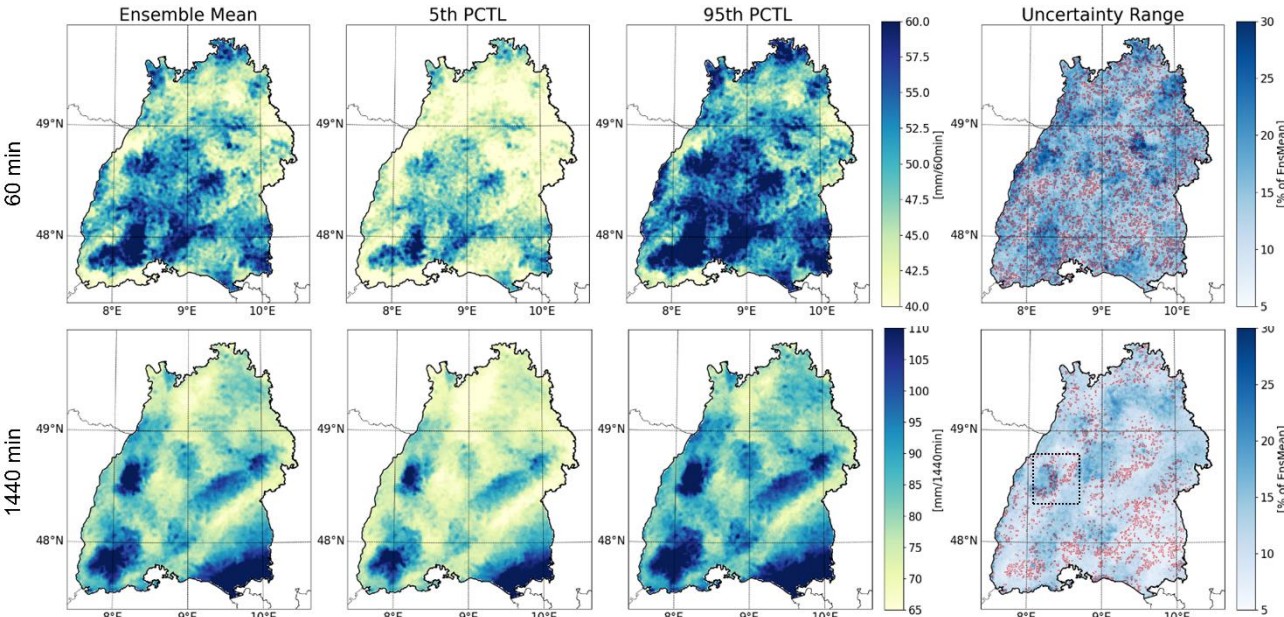

**Figure 6: Ensemble mean (left column) and the 5th and 95th percentiles (two middle columns) of a 100yr design storm based for two durations (60 minute events – upper row; 1440 minute events – bottom row). Additionally, the ensemble uncertainty range (difference between the 95th and the 5th percentile of the full (bootstrapping & sampling) 5000 member) is depicted (right column). Regions with a large (> 65% of the range) contribution of the sampling uncertainty are marked with red. The black dashed square in the panel in the lower right defines the northern Black Forest region discussed in the text.**


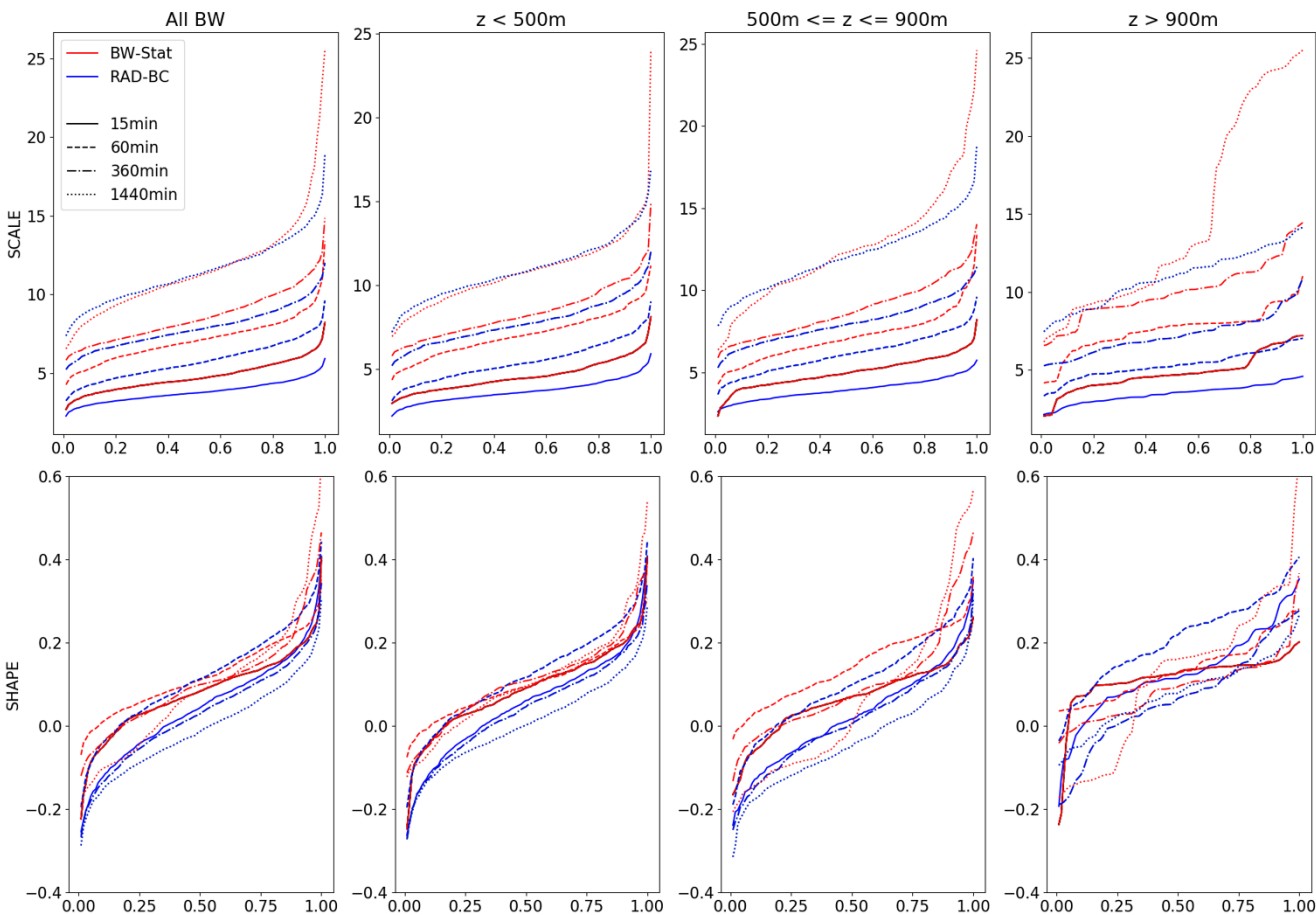

Figure 7: Spatial cumulative frequency distributions (CFD) of the scale (upper row) and shape (bottom row) parameter for the BW-Stat and RAD-BC datasets, when comparing stations and radar data at the location of all stations (left column) and for three different subsets filtered by the altitude of the respective station locations.

**Appendix**

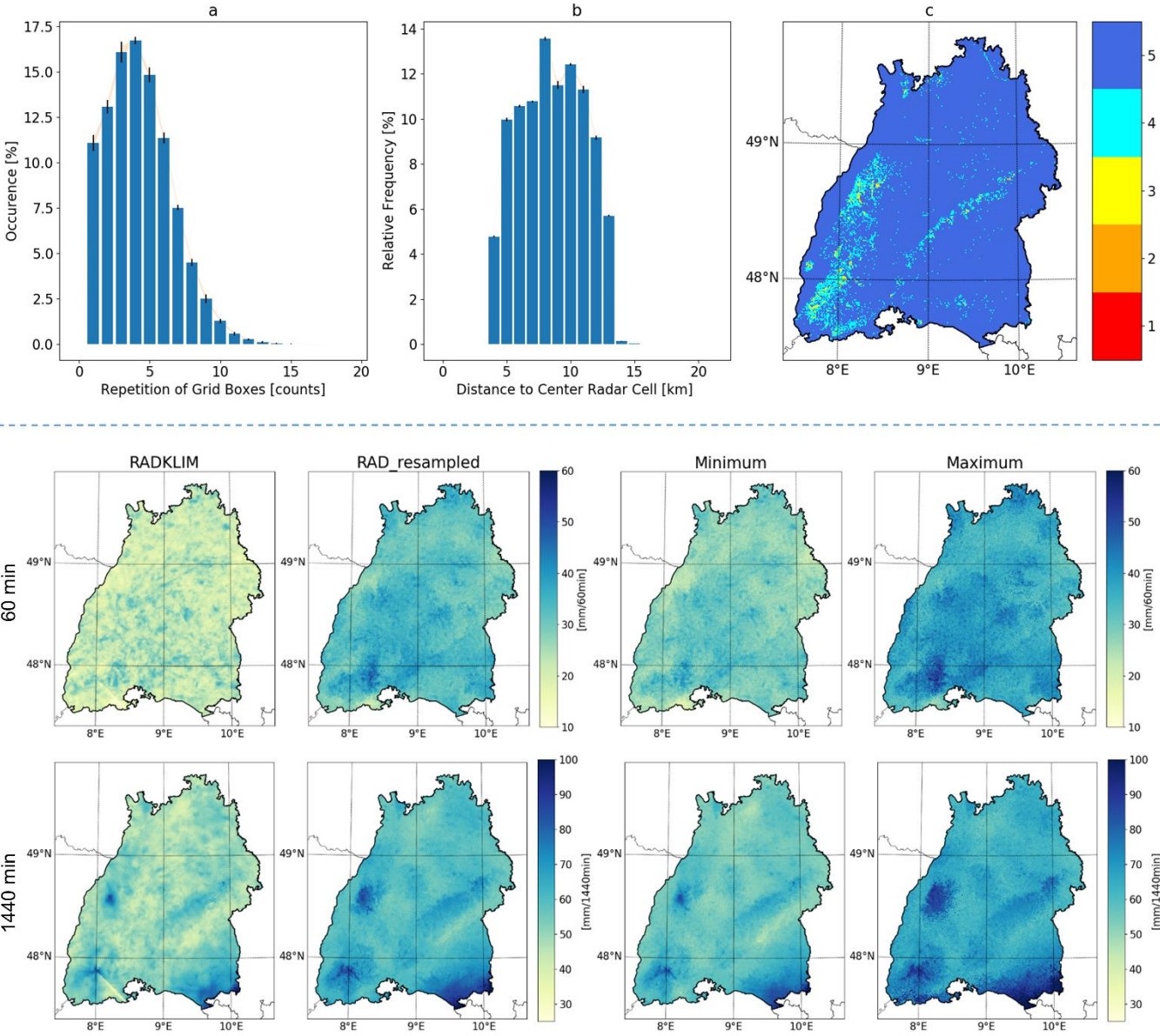

**Figure A1: Upper three panels: Distribution of the fraction of (a) cumulative occurrences of individual radar cells in the final sample; (b) the distance of the sampled radar cells to the cell of interest. (c) Spatial distribution of the effective ensemble size. The distributions in a and b are for a single ensemble member, while the error bars indicate the rather small variation among the five ensemble members. Lower Panels: Mean rainfall sum of the rank 1 to 10 events for two (60 & 1440 minutes) different event durations of the original RADKLIM data (left), and the resampled, but not corrected RADKLIM data (2nd left). The panels on the right hand side depict the minimum (2nd right) and maximum (right) of the ensemble generated through multiple variations of the sampling parameters.**

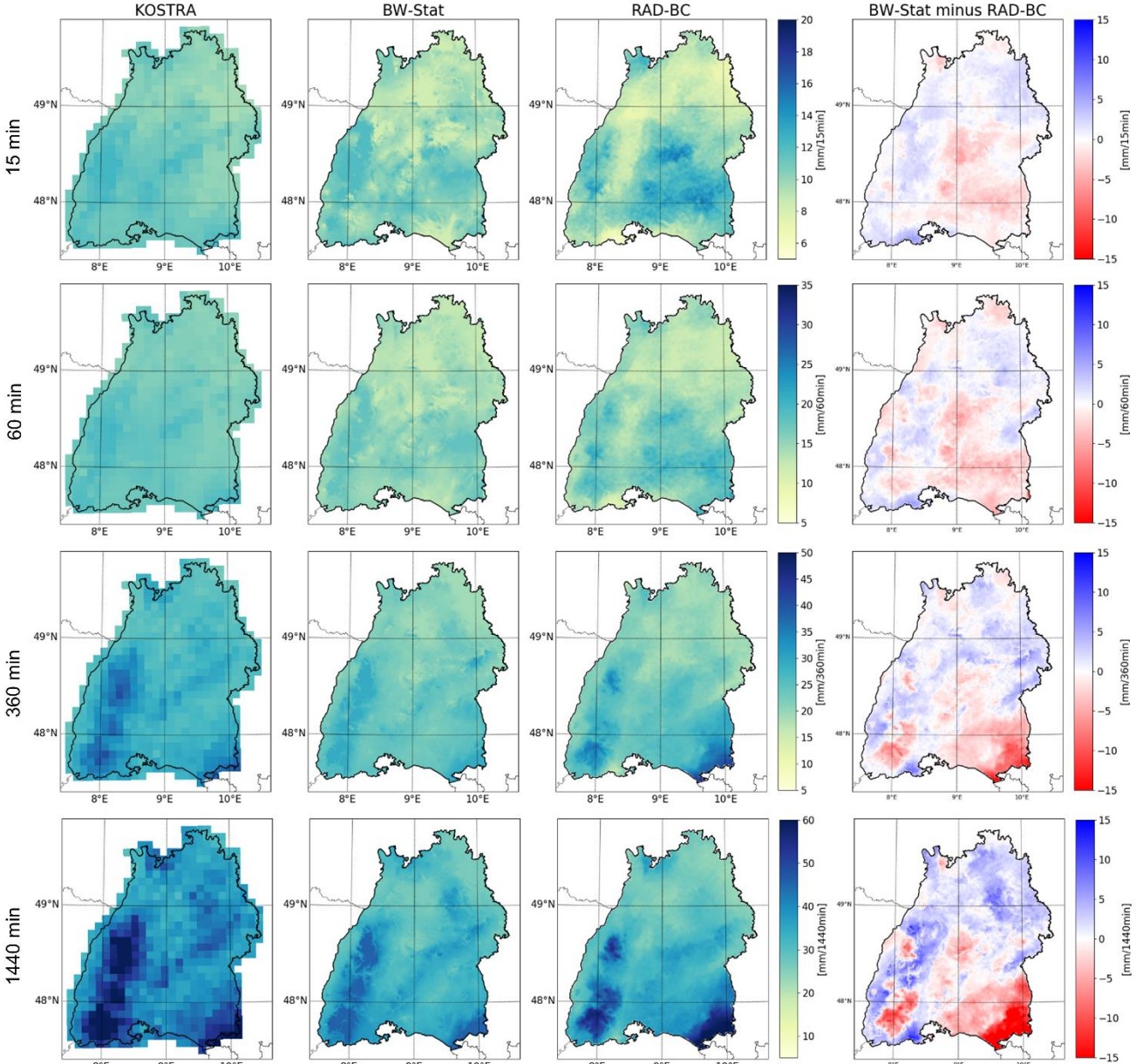

**Figure A2: Magnitude of 1yr design storms for four different event durations (15, 60, 360 and 1440 minutes, depicted in rows) and three different datasets (KOSTRA, BW-Stat, RAD-BC, depicted in columns). Additionally, the difference between BW-Stat and RAD-BC is depicted (right column). Note that in the BW-Stat dataset all values below/above the 5th/95th percentiles are set to the respective percentile value.**

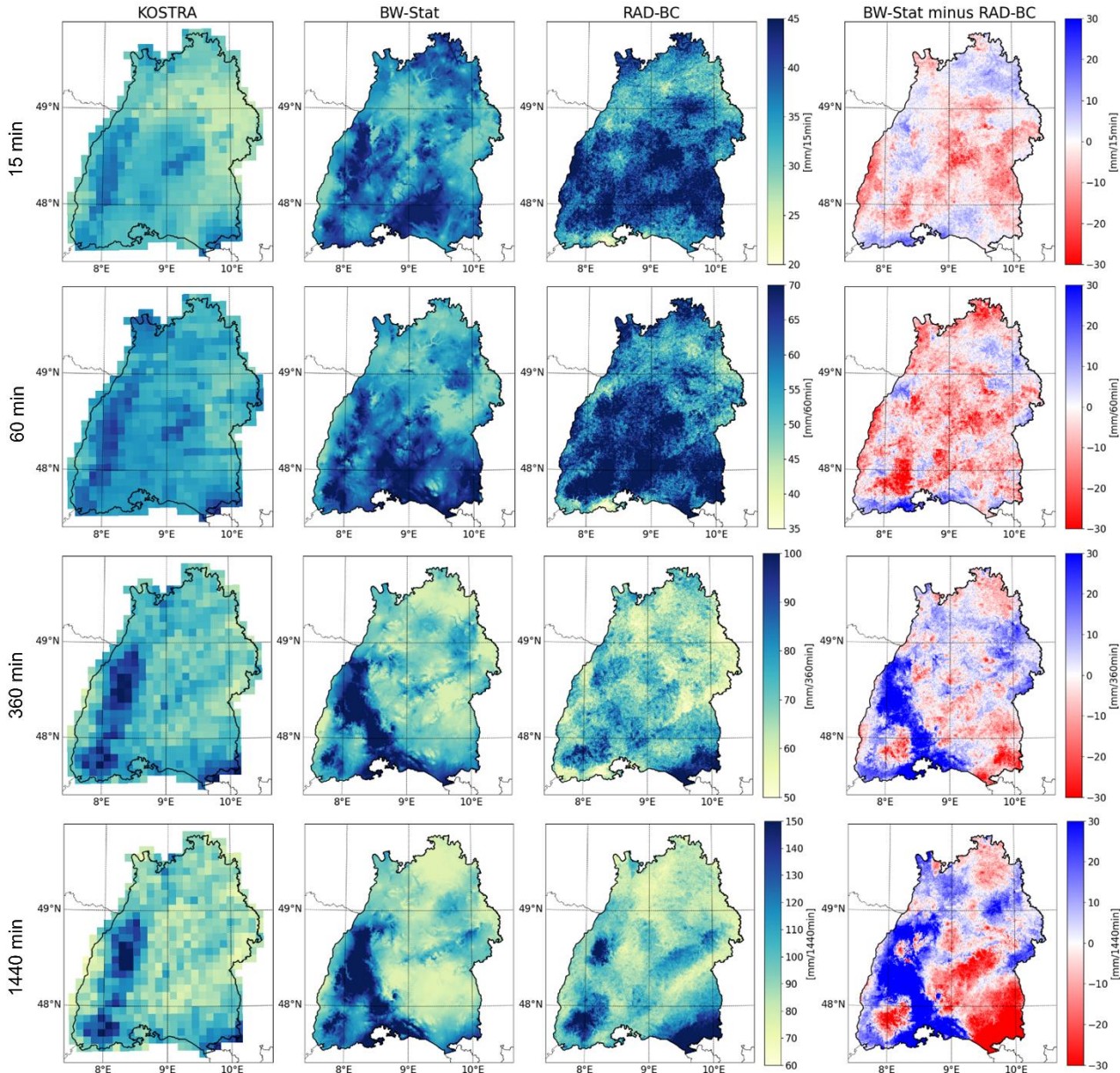

**Figure A3: Magnitude of design storms with a return rate of 100 years for four different event durations (15, 60, 360 and 1440 minutes, depicted in rows) and three different datasets (KOSTRA, BW-Stat, RAD-BC, depicted in columns). Additionally, the difference between BW-Stat and RAD-BC is depicted (right column). In contrast to Fig. 4, RAD-BC is correct using a multiplicative bias correction approach. Note that in the BW-Stat dataset all values below/above the 5th/95th percentiles are set to the respective percentile value.**