# Peer review of "Enhancing the usability of weather radar data for the statistical analysis of extreme precipitation events"

_Hydrology and Earth System Sciences, 2021_

## Referee Comment (RC1)

**"Enhancing the usability of weather radar data for the statistical analysis of extreme precipitation events"**

By  Andreas Hänsler and Markus Weiler

**Summary:**

A method to estimate spatially varying design storms with a return period of up to 100 years on the basis of short weather radar precipitation estimates and long but low spatial density gauge data is proposed. An example of application for the state of Baden Württemberg, Germany is presented. The main finding is that the spatial structure seems to be more realistic in the weather radar based product. However, despite the bias correction, the absolute magnitude of the radar-derived design storms remains too low, which should be addressed in future studies.

**Evaluation:**

In my opinion, the strong points of this study are:

a) Radar data are combined with gauge information to derive spatially varying design storms
b) Uncertainty is assessed through bootstrapping and ensembling.
c) The figures are nice.
d) The approach is potentially interesting.

The weaknesses are:

a) The English and the writing could be improved.
b) The paper is too long.
c) There are some parts in the methods section that need to be clarified.
d) The bias correction method used by the authors is controversial.
e) Serious doubts remain about the usefulness/reliability of the new dataset.
f) There are a lot of subjective parameters/choices (e.g., the way the radar grid cells are sampled). No proper sensitivity analysis is performed.

**Scientific significance:** Good (2)
**Scientific quality:** Fair (3)
**Presentation quality:** Good (2)
**Recommendation:** Reconsider after major review.

**Major Comments: (MC)**

**MC1 :** A lot of the methods are described in plain text, without the use of any equations or formal mathematical notations. As a result, it is hard to understand what exactly has been done to the data.

For example: ll.157-158 ``*In a final step, both spatial sampling probabilities are normalized with the respective maximum probability, added together and again normalized by the maximum to generate the final spatial probability distribution for the sampling (Fig. 1, panel bIII).''*

> Suggestion: Add some flowcharts or equations to complement the text and clarify the different steps, transformations or models used, especially in the methods section. Every step needs to be clearly described and reproducible.

**MC2:** The English is poor and the writing is overly complicated. The paper is too long for its content.

> Suggestion: substantially shorten the text by removing unnecessary words, expressions or sentences. Get rid of fillers such as "certainly", "basically", "already". Write shorter sentences and go to the point. Refrain from adding too much information in parentheses. Before including parentheses, check if they are essential.

**MC3:** It is still unclear whether RAD-BC can be used to build reliable regional design storms. Yes, the radar data are spatially more explicit, which removes the need for interpolation. But radar data have two huge disadvantages: 1) they are much shorter and 2) they are biased and more uncertain than gauge data. Given these drawbacks, I would argue that it's probably safer to build design storms based on a long rain gauge time series, even if the gauge is several hundreds of kilometers away and not very representative of the region of interest. But I might be wrong and the challenge is to find a good balance between having enough data for extreme value analysis and being spatially representative. So which approach is better, and under what circumstances? You could do more to answer this important question!

For example, spatial variations in small-scale extremes (15 min) are likely to be small, due to the random nature of convection. It's only at the daily aggregation time scales and beyond that the spatial structure really starts to matter (due to orographic enhancement).  Does this mean that RAD-BC only has value for longer duration extremes?

> Suggestion(s): Improve the validation part and clearly explain why/where you think your new RAD-BC dataset is valuable. Look for new ways to assess the effect of interpolation and bias correction. Look at differences in return values as a function of elevation and/or distance to gauges. Consider adding a case-study where you focus on specific locations/areas for which you know (from literature) that current methods provide bad estimates and compare them to what you get using RAD-BC. You need to convince the reader that RAD-BC offers real tangible advantages compared to simply interpolating design storms from gauges.

**MC4:** the method used to identify the peaks over threshold for fitting the GPD using partial series on ll.180-189 is not explained very well.

> Suggestion: Clarify Equation 1 by mentioning how the data from the 5 time series corresponding to the 5 radar cells were combined to estimate the GPD. Give a step-by-step description of the data extraction procedure, using equations and formal mathematical notations.

**MC5:** the quantile matching method you use for the location parameter in section 2.5 is questionable: In particular, I wonder why you decided to only bias-correct the location parameter but not the scale parameter as well, since the two are usually correlated. By only changing the location parameter, you are at risk of under/overestimating the spread of the GPD. An underestimated scale parameter might explain why RAD-BC has lower return values compared to BW-Stat, even after bias correction.

> Suggestion(s): Explain why you think it is justified to only adjust the location parameter during the bias correction. Explore what would happen if you adjust both the location and scale parameters simultaneously. If possible, explore alternatives for how to bias correct the radar-derived GPD parameters without depending too much on the 1-year return period.

**MC6:** The structure of the paper could be improved. For example, one could argue that section 3.1, which presents the sampling statistics, is not really a result (in the scientific sense) but merely a continuation of section 2.4.

> Suggestion: consider moving 3.1 to the methods section and merging it with 2.4 while shortening the text. There's nothing really surprising or valuable in this section that the reader can learn and all the statistics you present highly depend on the way you select the cells in the first place.

**Minor comments :**

- ll.127-128 : ``*It further has to be mentioned that in the final product all design rainfall values below/above the 5th/95th percentile have been set constant (to the 5th/95th percentile) in order to prevent for extremely low/high outliers.*''

> Not clear. Please reformulate. What quantiles are you referring to ? What do you mean by extremely low/high outliers ?

ll. 149-151 : ``*The maximum sampling radius was set to 25 km (cells). These numbers are chosen in order to reflect the typical size of a convective cell in Germany (~25 to 40 km for hourly events in the summer season in BaWu, Lengfeld et al., 2019).*''

> The actual size of a convective cell is much smaller. But because of the hourly aggregation time scale and cell motion, the impacted area is in the order of 25-40 km. Please reformulate the sentence to convey the right meaning.

ll.160-163 : ``*In order to prevent that neighbouring cells are sampled (which actually would limit the number of additional rainfall events), the sampling probabilities of the cells in a radius of 4 km (cells) of the cell are reduced below the threshold value after each sample is drawn (Fig. 1, pnel bIV).*''

> Not clear. Please reformulate the sentence to make sure the reader can understand and reproduce what you have done. Suggestion : first explain what you have done, then mention the rational behind it in a separate sentence/paragraph.

ll.272-273 : ``*Note that the RAD-BC dataset represents the ensemble mean of the five individual sample products and that the data is spatially smoothed with a 3 by 3 cell filter to avoid single outliers.*''

> What happens if you do not smooth? What are these «single outliers» exactly and how large are they?

ll.325-330: "*In order to reveal the uncertainty contribution resulting from the ensemble sampling we highlighted regions with a large (> 75% of the range) contribution of the sampling uncertainty. Generally, the contribution of the sampling uncertainty is larger in regions with a lower overall uncertainty range. However, there are various spots with relatively larger uncertainty that are dominated by the sub-sampling uncertainty. The previously mentioned enhanced uncertainty in the northern Black Forest case seems to be substantially influenced by sampling uncertainty in its eastern parts, [...]*"

> This is a good example of a passage that is difficult to read/understand and for which the writing could be improved.

ll. 348-349 : *``The lower values for the scale/shape parameters of RAD-BC can partly be attributed to the fact, that for high rainfall intensities radar data is known to underestimate rainfall amounts due to the  reflectivity bounds (e.g. Schleiss et al., 2020).''*

> No idea what you are referring to here. In the paper you cite, I show that radar rainfall estimates during heavy rain exhibit a strong (increasing) conditional bias with intensity. This bias persists despite the radar operators' best effort to combine/merge radar data and perform frequent bias correction (e.g., hourly mean field bias adjustments). It's likely that the German radar network suffers from the same problems. The fact that the bias increases with intensity means that a simple quantile mapping based on the 1-year return period probably won't be enough to reliably estimate longer return periods using radar. Note that the conditional bias is probably due to multiple factors, including signal attenuation, range effects, and natural variations in the raindrop size distributions with intensity, which affects the Z-R relationship (locally).

- ll.372-373 : *``A detailed comparison of the different bias correction approached and their implications on the derived design storms is currently in progress.''*

> This is a good example of a sentence that could be deleted to shorten the text.

**Typos :**

- ll. 125-127 : *``Especially over the mountain regions of the Black Forest, Swabian Jura and Alpine Foothills (see Fig. 1a) the horizontal distance in-between the can be up to 80km.''*

> A word seems to be missing here.

l.207-208 : *``This is usually triggered by the fact that radar measurements represent an integrated measurement of 1km x 1km while station data is a point measurement''*

> Triggered does not appear to be the right word here. Explained or caused would be better.

- l.239 : *``If two cells (e.g. COI and one of the four sampled cells)  are duplicated the effective ensemble size is set to four.''*

- l.290 : *``This can also be proofed with a spatial correlation analysis''*

> Bad English. Please reformulate.

- l.295 : *``Interestingly, the  spatial pattern in the BW-Stat dataset is following this behaviour.''*

- l.351 : *``The average rainfall amount per heavy rainfall day (< 20 mm of rainfall) is, however, almost identical (Kreklow, et al.,2020).''*

> The writing of this sentence could be improved. In addition, shouldn't a heavy rainfall day have > 20 mm of rainfall instead of < 20 mm ?

- l.367 : *``However, for BaWu there are only two stations with high temporal precipitation records available with a data series length of more than 50 years (Steinbrich et al., 2016)  posing a major challenge for this approach''*

- ll.369-372 : *``Another possible approach would be to base the correction on the  underlying observed rainfall events itself instead of correcting design storms. This would have the*

*benefit that  the high-intensity events would be directly corrected and  not derived based on the correction factors estimated for less-intense events.''*

*- ll.381-384 : ``The established method has various positive aspects. The main improvement is the development of a spatially homogeneous dataset that allows for the calculation of  extreme events that is not dependent on spatial interpolation methods that is often the main error source when building a regional dataset based on station data. Moreover, the chosen sampling approach allows  to control the sampling region based on physical aspects while preventing  artificial circular structures previously reported in literature (e.g.  Goudenhoofdt et al., 2017) ''*

---

## Author Comment (AC3)

**Review of „Enhancing the usability of weather radar data for the statistical analysis of extreme precipitation events"**

Dear Anonymous Reviewer,
thank you very much for your very detailed review of our manuscript and your very valuable suggestions to improve it.
Please find our response to the various points you raised below (in red).
Best regards,
Andreas Hänsler and Markus Weiler
* * *
**General comments:**

The authors provide a quite novel approach to estimate design rainfall from weather radar in Baden-Württemberg (BW). The main idea is the pooling of data from radar pixels in the proximity of the target cell to increase the sample size beyond the 19 years available record length. The radar data are bias corrected and compared against two station-based data sets, the German design storm standard KOSTRA and a regional data set from BW.

Although the approach is quite heuristic with several arbitrary assumptions and decisions (e.g., search radius, local estimation method, interpolation of parameters over durations, independence assumptions, etc.) it is practically pragmatic and statistically satisfactory. There are some major issues which need further attention and discussion. The first is the selection of events within a small search radius with the assumption of spatial independence. Second, the many arbitrary assumptions need to be better justified. Third, there is a need for better and more formal description of the methods.

Altogether the idea is good, the results are interesting and plausible. The text is significant and reads quite well although the English could be improved. I would recommend publication after major revision.

**Specific comments:**

1. Title: The title suggests a postprocessing of radar data for a later statistical analysis. However, you have done already the analyses. I would recommend to adjust the title e.g. something like "A pooling approach for design storm estimation using weather radar data – a case study for BW"

   Thank you very much for this suggestion. However, we would rather like to keep the title since we want to focus rather on the methodological aspects that are needed to post-process rainfall radar to be used for statistical extreme value analysis than on the results of the EVA itself. However, we probably have to make this more clear in our manuscript.

2. Lines 31: The non-stationarity is not considered in the approach. Of course, with only 19 years of observations this is hardly feasible. However, at least a brief discussion or an outlook should be included.

   Thank you very much for pointing this out – we will include this in a revised version of the manuscript.

3. Line 89: "Reassembling via running sums" becomes not clear. Usually, the highest temporal resolution of 5 min is used to build the extreme values series for all durations by calculating sums over a moving window with width equal to the duration and moving step of 5 min? Is that the procedure used here?

    Yes, exactly that is what has been done. We will add some flow charts (suggestion of Reviewer 1) to improve the description of the sampling and data processing.

4. Lines 146 ff: I think I could finally figure out how the sampling locations are selected but the description is weak. Please, reformulate and explain better. There are several arbitrary assumptions: why have you selected the normal distribution, how did you define its parameters, how did you select the 0.8 threshold, etc.? These need to be justified and discussed.

    Yes, we agree that this should be improved. We will also add a flow chart on this to improve the description of the sampling and data processing. The choice of parameters defining the boundaries is indeed somehow subjective. They were basically chosen in a way that we give the sampling process a high degree of freedom adapted to the specific local conditions (so no fixed sampling area like circles or other structures have been prescribed). On the other hand, the parameter choice takes care of the fact that only cells are sampled that are somehow regionally representative for the COI. The resulting combination of probability distribution and threshold gave the best results (see Fig 2), but others were tested.

5. Line 157: Include equation for normalisation.

    We will add this to the flow chart

6. Line 168: The "sub-sampling is not adapted for different event durations"; I guess you mean by that, that the same locations have been sampled independent of durations?

    Correct, that is what we meant. We will adapt the sentence.

7. Line 189: What about spatial independence? Is this minimum separation time of 48 hours between events applied on the whole compiled data set from all 5 locations together? Only that way a spatial independence can be assumed. On the other hand, in that case considering the small search radius I would assume, that the sample from the five locations is not really comparable with a real 100-year sample; it probably will contain less extreme events and finally lead to an underestimation, which partly may explain the results.

    Yes, for the final dataset of resampled events we applied the 48h criteria.
    The search radius is a compromise between the spatial representativeness of the COI and the inclusion of additional extreme events. We find that through the sampling process a significant increase in the rainfall amount of the top events could be reached. But we also believe that the general bias of the RAD-BC events would still remain, even we would enlarge the larger sampling radius. This is mainly due to the known underestimation of high intensity rainfall events in the radar.

8. Line 197 ff: In the independent fitting of distributions for different durations order relations problems may occur. This is accounted for in DWA (2012) by smoothing the parameters over the durations, which is a bit "old-fashion". Please, explain more in detail which method has been applied here and discuss also alternatives.

So far, we have not applied any smoothing across the different event durations. This was done to be in close agreement with the method used to generate the BW-Stat dataset. Also, the focus of the paper is more on the combined pooling process and bias-correction as well as the resulting changes in spatial patterns. But of course we could briefly discuss the cross-duration relation (at least briefly in the discussion section) although a parameter smoothing might impact the positive effects the approach has on the varying spatial patterns of design rainfall events for different durations.

9. Line 217 ff: Please provide equations for the quantile mapping approach.

We will include the equation in the respective flow chart

10.       Fig. 3: Are the probability distributions compiled from all stations/locations together? If yes, how many stations are included?

They are calculated for all grid boxes. BW-Stat and RAD-BC have the same number of grid boxes since they are on the same grid. KOSTRA has a substantially lower number of grid boxes (hence the pdf is less smooth). We will include this information in the figure caption.

11.       Line 290: What is meant by spatial correlation analyses? Do you refer to correlations between rainfall and elevation? If yes, this is a cross-correlation but not a spatial correlation, which is usually used to quantify spatial persistence by correlation-distance relationships like the variogram, which by the why could have been employed for a more objective selection of the neighbourhood for sampling.

It is actually the cross correlation between the spatial patterns of REGNIE precipitation and the RAD-BC data. We will change this in the manuscript.

Regarding the variogram, we agree that this could also have been a valuable approach to identify a potential sampling area. However, also with this approach some subjective assumptions have to be made, since the correlation will be highest closest to the COI. But we want to sample in certain boundaries (linked to the size of convective cells). Based on the suggestions of Reviewer 1 we will move the figure describing the sampling statistics into the attachment. But we can add some comparable analysis in the respective figure on how the sampling statistics would change, if a variogram approach would be used.

12.       Fig. 5: Same question as for Fig. 3.

see reply above

13.       Line 315: Why are you using an 80% confidence interval here; usually a 90% interval between 5% and 95% quantiles is used?

Thank you very much for pointing this out. We will change it to 5/95%.

14.       Discussion/ conclusion: The new product has been compared against 2 reference data sets, but no strict validation has been carried out as usually desired. This is of course difficult since the truth is not known. However, often the long-term observations (>30 years) are applied as truth in a cross-validation mode. The application of this is also difficult here since the RADKLIM data set itself is a merged product involving these stations which makes this infeasible. At least a discussion of this problematic is required and optimal would be some suggestions for further research.

Thanks for pointing this out. Since the methodological differences and the much lower spatial resolution of KOSTRA a one-to-one validation with KOSTRA is not possible. And the BW-Stat dataset is actually a pooled dataset itself.

Furthermore, we already know that both station based datasets have deficits in their spatial patterns caused by (i.) the limited number of stations included and (ii) the explicit consideration of the topography in the interpolation, which is (at least for short duration events) somehow questionable. We mention this problem of not having a clear validation dataset already in the manuscript, but we will make sure that it is better reflected in the discussion of the results.

---

## Author Response (AR1)

**"Enhancing the usability of weather radar data for the statistical analysis of extreme precipitation events"**
By Andreas Hänsler and Markus Weiler

Dear Marc Schleiss,
thank you very much for your very detailed review of our manuscript and your very valuable suggestions to improve it.
Please find our response to the various points you raised below (in red).
Best regards,
Andreas Hänsler and Markus Weiler
* * *
**Summary:**
A method to estimate spatially varying design storms with a return period of up to 100 years on the basis of short weather radar precipitation estimates and long but low spatial density gauge data is proposed. An example of application for the state of Baden Württemberg, Germany is presented. The main finding is that the spatial structure seems to be more realistic in the weather radar based product. However, despite the bias correction, the absolute magnitude of the radar-derived design storms remains too low, which should be addressed in future studies.

**Evaluation:**
In my opinion, the strong points of this study are:
a) Radar data are combined with gauge information to derive spatially varying design storms
b) Uncertainty is assessed through bootstrapping and ensembling.
c) The figures are nice.
d) The approach is potentially interesting.

The weaknesses are:
a) The English and the writing could be improved.
b) The paper is too long.
c) There are some parts in the methods section that need to be clarified.
d) The bias correction method used by the authors is controversial.
e) Serious doubts remain about the usefulness/reliability of the new dataset.
f) There are a lot of subjective parameters/choices (e.g., the way the radar grid cells are sampled). No proper sensitivity analysis is performed.

**Scientific significance:** Good (2)
**Scientific quality:** Fair (3)
**Presentation quality:** Good (2)
**Recommendation:** Reconsider after major review.

**Major Comments: (MC)**

**MC1:** A lot of the methods are described in plain text, without the use of any equations or formal mathematical notations. As a result, it is hard to understand what exactly has been done to the data. For example: ll.157-158 ``In a final step, both spatial sampling probabilities are normalized with the respective maximum probability, added together and again normalized by the maximum to generate the final spatial probability distribution for the sampling (Fig. 1, panel bIII).''
> Suggestion: Add some flowcharts or equations to complement the text and clarify the different steps, transformations or models used, especially in the methods section. Every step needs to be clearly described and reproducible.

Thank you very much for this very valuable suggestion. Also the other reviewers pointed out that the method section still needs some improvement. So we added a flow chart to give a step by step overview on the data processing (new Fig 2). We also completely rewrote the method section and included also the respective equations in the script. We hope that this clarifies the sampling process as well as the data analysis.

**MC2:** The English is poor and the writing is overly complicated. The paper is too long for its content. Suggestion: substantially shorten the text by removing unnecessary words, expressions or sentences. Get rid of fillers such as "certainly", "basically", "already". Write shorter sentences and go to the point. Refrain from adding too much information in parentheses. Before including parentheses, check if they are essential.

Also the other reviewers pointed out that the language should be improved. We carefully rechecked what we have written and shortened some of the text passages (e.g. we completely removed the paragraph on the sampling results). We also cleaned the paper from filler words as much as possible. But since all reviewers asked for a more detailed description of the sampling process, more details on the station data as well as adding more points into the discussion like e.g. the parameter choice and related uncertainties we unfortunately failed to shorten the overall paper length but in fact actually have a slightly longer paper than before.

**MC3:** It is still unclear whether RAD-BC can be used to build reliable regional design storms. Yes, the radar data are spatially more explicit, which removes the need for interpolation. But radar data have two huge disadvantages: 1) they are much shorter and 2) they are biased and more uncertain than gauge data. Given these drawbacks, I would argue that it's probably safer to build design storms based on a long rain gauge time series, even if the gauge is several hundreds of kilometers away and not very representative of the region of interest. But I might be wrong and the challenge is to find a good balance between having enough data for extreme value analysis and being spatially representative. So which approach is better, and under what circumstances? You could do more to answer this important question! For example, spatial variations in small-scale extremes (15 min) are likely to be small, due to the random nature of convection. It's only at the daily aggregation time scales and beyond that the spatial structure really starts to matter (due to orographic enhancement). Does this mean that RAD-BC only has value for longer duration extremes?

This is actually an example where our dataset delivers an added value, but we obviously have to make it more clear in the manuscript. Both station based products that are included for comparison, show some spatial patterns linked to topography already for the short event durations. This is not surprising when considering the fact that the topography was explicitly included in the spatial interpolation of the data (and in the case of the BW-Stat dataset, additionally in the selection of stations to extend the data series). So we believe that especially for shorter durations, our dataset provides an added value (at least in the spatial patterns).

Suggestion(s): Improve the validation part and clearly explain why/where you think your new RAD-BC dataset is valuable. Look for new ways to assess the effect of interpolation and bias correction. Look at differences in return values as a function of elevation and/or distance to gauges. Consider adding a case-study where you focus on specific locations/areas for which you know (from literature) that current methods provide bad estimates and compare them to what you get using RAD-BC. You need to convince the reader that RAD-BC offers real tangible advantages compared to simply interpolating design storms from gauges.

Thank you very much for the suggestions. Given what we wrote above, we clearly think that there is an added value in the spatial representation of design storms when using radar data. An additional

information we now provide, is to compare our remaining bias with the interpolation error for the BW-Stat data, since we had the possibility to redo the spatial interpolation of the station based product and therefore could compare the potential interpolation error at each station location. We find that the interpolation error of the station data is quite large itself. We also highlighted the fact that events from single stations influence the absolute amounts and spatial patterns over relatively large areas.

**MC4:** the method used to identify the peaks over threshold for fitting the GPD using partial series on ll.180-189 is not explained very well.

> Suggestion: Clarify Equation 1 by mentioning how the data from the 5 time series corresponding to the 5 radar cells were combined to estimate the GPD. Give a step-by-step description of the data extraction procedure, using equations and formal mathematical notations.

Thank you very much for the valuable suggestion. We included this information now in the revised version (see also response to first comment).

**MC5:** the quantile matching method you use for the location parameter in section 2.5 is questionable: In particular, I wonder why you decided to only bias-correct the location parameter but not the scale parameter as well, since the two are usually correlated. By only changing the location parameter, you are at risk of under/overestimating the spread of the GPD. An underestimated scale parameter might explain why RAD-BC has lower return values compared to BW-Stat, even after bias correction.

> Suggestion(s): Explain why you think it is justified to only adjust the location parameter during the bias correction. Explore what would happen if you adjust both the location and scale parameters simultaneously. If possible, explore alternatives for how to bias correct the radar-derived GPD parameters without depending too much on the 1-year return period.

Thank you very much for the valuable suggestions. In the manuscript we describe two BC options, both based on the 1-year return period. By using a multiplicative approach, we indirectly adapted also the scale and shape parameter, since we multiply all events with a certain factor leading to a larger absolute precipitation amount added to the largest events.
We believe that an explicit correction of the scale parameter is possible, however, given the nature of the station data (most of them are of rather short duration of ~20yrs and have been resampled itself), we think that an explicit correction of the scale factor is not justified. We tried to make this clearer in the manuscript, why we think that it is not justified to also correct the scale parameter directly. Looking at figure 7 we also conclude in the paper that a correction of the scale parameter could lead to a smaller remaining bias for the shorter durations, but for the long duration events it might be the shape parameter that dominates. Furthermore, Fig7 reveals some strange behavior of the scale parameter over the higher elevated areas of the analysed domain which is not seen in the location parameter and is most likely linked to the regional-subsampling process applied in the BW-stat data. This is another reason why we think we so far did not include other parameters into the bias correction.
We are meanwhile working on a more complex BC-correction method based on the individual events, which looks promising but still has some major drawbacks and needs some more testing – but this will probably be a study for itself.

**MC6:** The structure of the paper could be improved. For example, one could argue that section 3.1, which presents the sampling statistics, is not really a result (in the scientific sense) but merely a continuation of section 2.4.

> Suggestion: consider moving 3.1 to the methods section and merging it with 2.4 while shortening the text. There's nothing really surprising or valuable in this section that the reader

can learn and all the statistics you present highly depend on the way you select the cells in the first place.

Thank you very much for this suggestion that we will follow. Actually section 3.1. was intended to proof that the spatial sampling is doing what it was expected to do, even we leave some degree of freedom. Applying an ensemble sampling approach could theoretically lead to a repeated sampling of the same cells/events, which would bias the result. Nevertheless, we agree that the respective figure is not a key element of the manuscript. Therefore, we actually moved the respective figure to the attachment section, and also kept only some key text elements in the method section 2.3.2

**Minor comments :**

ll.127-128 : ``It further has to be mentioned that in the final product all design rainfall values below/above the 5th/95th percentile have been set constant (to the 5th/95th percentile) in order to prevent for extremely low/high outliers.'' Not clear. Please reformulate. What quantiles are you referring to? What do you mean by extremely low/high outliers?

This sentence actually refers to the BW-Stat dataset, were all values below/above the 5th/95th percentile (spatially) were capped. We can reformulate this, but it was not applied to our radar-based dataset and it was not done by us.

ll. 149-151 : ``The maximum sampling radius was set to 25 km (cells). These numbers are chosen in order to reflect the typical size of a convective cell in Germany (~25 to 40 km for hourly events in the summer season in BaWu, Lengfeld et al., 2019).'' The actual size of a convective cell is much smaller. But because of the hourly aggregation time scale and cell motion, the impacted area is in the order of 25-40 km. Please reformulate the sentence to convey the right meaning.

The full section was reformulated

ll.160-163 : ``In order to prevent that neighboring cells are sampled (which actually would limit the number of additional rainfall events), the sampling probabilities of the cells in a radius of 4 km (cells) of the cell are reduced below the threshold value after each sample is drawn (Fig. 1, pnel bIV).'' Not clear. Please reformulate the sentence to make sure the reader can understand and reproduce what you have done. Suggestion: first explain what you have done, then mention the rationale behind it in a separate sentence/paragraph.

The full section was reformulated

ll.272-273 : ``Note that the RAD-BC dataset represents the ensemble mean of the five individual sample products and that the data is spatially smoothed with a 3 by 3 cell filter to avoid single outliers.'' What happens if you do not smooth? What are these «single outliers» exactly and how large are they?

They are basically rather small (see figure below). We just smoothed it since the KOSTRA and BW-Stat datasets are highly smoothed due to the interpolation.

[Figure]

ll.325-330: "In order to reveal the uncertainty contribution resulting from the ensemble sampling we highlighted regions with a large (> 75% of the range) contribution of the sampling uncertainty. Generally, the contribution of the sampling uncertainty is larger in regions with a lower overall uncertainty range. However, there are various spots with relatively larger uncertainty that are dominated by the sub-sampling uncertainty. The previously mentioned enhanced uncertainty in the northern Black Forest case seems to be substantially influenced by sampling uncertainty in its eastern parts, [...]"

> This is a good example of a passage that is difficult to read/understand and for which the writing could be improved.

We refomulated this.

l. 348-349 : ``The lower values for the scale/shape parameters of RAD-BC can partly be attributed to the fact, that for high rainfall intensities radar data is known to underestimate rainfall amounts due to the  reflectivity bounds (e.g. Schleiss et al., 2020)."

> No idea what you are referring to here. In the paper you cite, I show that radar rainfall estimates during heavy rain exhibit a strong (increasing) conditional bias with intensity. This bias persists despite the radar operators' best effort to combine/merge radar data and perform frequent bias correction (e.g., hourly mean field bias adjustments). It's likely that the German radar network suffers from the same problems. The fact that the bias increases with intensity means that a simple quantile mapping based on the 1-year return period probably won't be enough to reliably estimate longer return periods using radar. Note that the conditional bias is probably due to multiple factors, including signal attenuation, range effects, and natural

variations in the raindrop size distributions with intensity, which affects the Z-R relationship (locally).

We actually wanted to point out the use of a fixed Z-R relationship in the radar data. We reformulated this.

- ll.372-373 : ``A detailed comparison of the different bias correction approached and their implications on the derived design storms is currently in progress.'' This is a good example of a sentence that could be deleted to shorten the text.

We removed this sentence

**Typos :**
Sorry for the typos and thank you very much for pointing them out. Of course we corrected them for all cases still valid after the revision of the manuscript.

**Review of „Enhancing the usability of weather radar data for the statistical analysis of extreme precipitation events"**

Dear Anonymous Reviewer,
thank you very much for your very detailed review of our manuscript and your very valuable suggestions to improve it.
Please find our response to the various points you raised below (in red).
Best regards,
Andreas Hänsler and Markus Weiler
* * *
The manuscript deals with the important and timely topic of determining design storms with return periods of up to 100 years from rather short time series of precipitation data from radar observations. The authors present a method to statistically extend time series of weather radar rainfall estimates by combining regional frequency analyses with subsequent bias correction. The results show improvement over the sampling approach by Goudenhoofdt et al. (2017) that is used as basis for their method, but uncertainties, e.g. a bias in the radar data for design storms with large return periods, still remain.

The study fits in the scope of HESS and is of interest to the research community. I have already reviewed an earlier version of the manuscript and the authors have taken some of my suggestions into account. However, some major concerns remain and new questions came up, that I listed below. I recommend major revisions of the manuscript before publishing it. I'd be happy to discuss my suggestions with the authors in the open discussion and clear up possible misunderstandings.

**Major comments:**

1. A major concern is the minimum distance of the radar cells that are considered to statistically extend the time series of the cell of interest. As far as I understand the cells have to be at least 4 km apart. The authors mention that the typical size of a convective cell in Germany is 40 km for hourly events according to Lengfeld et al. 2019 (p.5, l.150 in this manuscript). Therefore, the minimum distance of 4 km seems a bit too small to me, especially when considering also daily events that have a much larger typical spatial extent. The authors mention that more or less the same amount of events have been taken from all pixels, but did the authors perform any kind of independence check for the time series from the cells that are combined to a long time series, e.g. the correlation of the time series?

   It is true that we sample rather close to the COI, in order to mainly sample cells that have similar rainfall characteristics. As shown in Figure 2b, the majority of sampled cells are in a distance range between 8 and 12 km. The events of the sampled cells will definitely have a certain amount of correlation (actually it is intended that they have) to the events in the COI - especially for the longer durations. However, since we have as prerequisite that single events (independent of the cell they are sampled from) have to be at least two days apart, we assume that we can ignore the autocorrelation effect in the EVA, since the duration of a single event is much shorter. So, based on this approach, all selected events for a raster cell are independent events, which are necessary for the extreme value analysis.

2. The aim of the study is to determine design storms with a return period of 100 years. Therefore, a method to extend rather short time series (19 years) from radar data by using additional data from similar regions is presented and compared to a station-based interpolate product. It makes sense to have a time series of more or less the length of the return period for the radar data. Therefore, a length of 95 years has been chosen which equals a combination of 5 pixels. But there is no information on the length of the station-based products that are used as references here. To my knowledge KOSTRA is contains 60 years of data. How reliable are the estimations of design storms with a return period of 100 years from KOSTRA? How many years of data are included in BW-stat? I was wondering how fair the comparison is when using data sets with different lengths. It would be beneficial to the manuscript if the authors add some information about this.

Most of the stations included in BW-Stat have actually a similar length than the radar data series and only a few are more than 40 yrs. Due to the short data series available, a spatial pooling approach was applied to construct the BW-Stat dataset, as well. We included the information on the length of the data series (also for KOSTRA) in the changed manuscript.

3. Although the authors extended Section 2.2 about the reference data sets some information are still missing (e.g. how many stations are considered, length of the time series, interpolation methods, etc.) A more detailed description of methods used in BW-stats and KOSTRA as well as the differences in the statistical approaches to determine design storms from those data sets is also desirable. The method for BW-Stat is briefly described in section 2.4. Maybe it would be better to have a general section about the methods first and then describe the data sets and their differences. Important information about the methods are missing that are crucial to understand the results and differences between the datasets.

We added more information on the method behind the two station based datasets. However, we also have been asked by Reviewer 1 to substantially shorten the manuscript, so we kept it short and refer to the respective reports. Considering the restructuring of data and method section we believe that we need to first describe the radar dataset before we can discuss the methods. Hence, we would prefer to leave the order of data and method description in its current state.

4. A more detailed description of the sampling process, the generation of the ensemble members, the bootstrapping method and the bias correction is needed to allow for better understanding of the results and of the choices made by the authors.

Reviewer 1 suggested to include some flow charts to make the sampling processes more clear. This suggestion we followed. Additionally, we completely restructured and rewrote the method section and included the respective equations.

5. Some findings are mentioned in the result section, but not sufficiently discussed in the discussion section. E.g. Why the spatial pattern in BW-stat is following the behaviour of RAD-BC for a return

period of 1 year (p.10, l.295). Extending the discussion section and analysing the results in more depth is necessary to enhance the quality of the manuscript.

Thanks a lot for pointing this out. We worked on the discussion of the results and also add some more analyses regarding the uncertainty of our radar-based data set with respect to the underlying sampling parameters as well as when compared to the interpolation error of the BW-Stat dataset.

The difference in the 1yr and 100yr pattern can actually be linked to the spatial resampling that was done in the BW-Stat dataset as well. Since stations in similar altitudes were preferably sampled a lot of the top events (especially over the more mountainous regions of BW) were from the same stations. We described this also more in detail in the updated manuscript.

**Minor comments:**

p.1, l.6-8: This sentence sounds odd to me. Please rephrase.

We tried to make it more clear.

p.1, l.17: A bracket seems to be missing here.

The reviewer is right. We changed it accordingly.

p.2, l.49: times series --> time series

The reviewer is right. We changed it accordingly.

p.2, l.62: …might not sufficient… --> …might not be sufficient…

The reviewer is right. We changed it accordingly.

p.4, l.107: What exactly are the methodological differences the authors mention here?

They are basically (i) the use of a different extreme value distribution (2 parameter GEV vs 3 parameter GPD) as well as (ii) that for some durations, the design storms in KOSTRA are interpolated from design storms of neighboring durations, while we calculate them explicitly for each of the durations. We elaborate more on this in the changed discussion section,

p.4, l.126: A word seems to be missing in this sentence.

The reviewer is right. The word 'station' is missing here

p.5, l.127: Is the limitation to values between the 5$^{th}$ and 95$^{th}$ percentile really necessary? How large the outliers? Please justify this decision.

This decision was not made by us, but the developers of the BW-Stat dataset. We tried to make this more clear.

p.12, l.371: …the also the… --> ….also the…

The sentence was rephrased as suggested by reviewer 1.

p.12, l.372: approached --> approaches

The sentence was deleted as suggested by reviewer 1.

**Review of „Enhancing the usability of weather radar data for the statistical analysis of extreme precipitation events"**

Dear Anonymous Reviewer,
thank you very much for your very detailed review of our manuscript and your very valuable suggestions to improve it.
Please find our response to the various points you raised below (in red).
Best regards,
Andreas Hänsler and Markus Weiler

**General comments:**

The authors provide a quite novel approach to estimate design rainfall from weather radar in Baden-Württemberg (BW). The main idea is the pooling of data from radar pixels in the proximity of the target cell to increase the sample size beyond the 19 years available record length. The radar data are bias corrected and compared against two station-based data sets, the German design storm standard KOSTRA and a regional data set from BW.

Although the approach is quite heuristic with several arbitrary assumptions and decisions (e.g., search radius, local estimation method, interpolation of parameters over durations, independence assumptions, etc.) it is practically pragmatic and statistically satisfactory. There are some major issues which need further attention and discussion. The first is the selection of events within a small search radius with the assumption of spatial independence. Second, the many arbitrary assumptions need to be better justified. Third, there is a need for better and more formal description of the methods.

Altogether the idea is good, the results are interesting and plausible. The text is significant and reads quite well although the English could be improved. I would recommend publication after major revision.

**Specific comments:**

1. Title: The title suggests a postprocessing of radar data for a later statistical analysis. However, you have done already the analyses. I would recommend to adjust the title e.g. something like "A pooling approach for design storm estimation using weather radar data – a case study for BW"

   Thank you very much for this suggestion. However, we would rather like to keep the title since we want to focus rather on the methodological aspects that are needed to post-process rainfall radar to be used for statistical extreme value analysis than on the results of the EVA itself. We tried to make this more clear in the updated version of the manuscript.

2. Lines 31: The non-stationarity is not considered in the approach. Of course, with only 19 years of observations this is hardly feasible. However, at least a brief discussion or an outlook should be included.

   Thank you very much for pointing this out – we tackled this aspect in the changed discussion section of the manuscript.

3. Line 89: "Reassembling via running sums" becomes not clear. Usually, the highest temporal resolution of 5 min is used to build the extreme values series for all durations by calculating sums over a moving window with width equal to the duration and moving step of 5 min? Is that the procedure used here?

> Yes, exactly that is what has been done. We added a flow chart (suggestion of Reviewer 1) summarizing the data processing and completely restructured and rewrote the method section to improve the description of the sampling and data processing.

4. Lines 146 ff: I think I could finally figure out how the sampling locations are selected but the description is weak. Please, reformulate and explain better. There are several arbitrary assumptions: why have you selected the normal distribution, how did you define its parameters, how did you select the 0.8 threshold, etc.? These need to be justified and discussed.

> Yes, we agree that this needed to be improved. We tried to describe it better in the changed method section.
>
> The choice of parameters defining the boundaries is indeed somehow subjective. They were basically chosen in a way that we give the sampling process a high degree of freedom adapted to the specific local conditions (so no fixed sampling area like circles or other structures have been prescribed). On the other hand, the parameter choice takes care of the fact that only cells are sampled that are somehow regionally representative for the COI. The resulting combination of probability distribution and threshold gave the best results (see Fig A1), but others were tested. We discuss this now in much more detail in the manuscript, also showing the results of the sensitivity studies in Fig A1.

5. Line 157: Include equation for normalisation.

> We included it

6. Line 168: The "sub-sampling is not adapted for different event durations"; I guess you mean by that, that the same locations have been sampled independent of durations?

> Correct, that is what we meant. We slightly adapted the sentence.

7. Line 189: What about spatial independence? Is this minimum separation time of 48 hours between events applied on the whole compiled data set from all 5 locations together? Only that way a spatial independence can be assumed. On the other hand, in that case considering the small search radius I would assume, that the sample from the five locations is not really comparable with a real 100-year sample; it probably will contain less extreme events and finally lead to an underestimation, which partly may explain the results.

> Yes, for the final dataset of resampled events we applied the 48h criteria.  We hope that this now becomes more clear in the revised version.
>
> The search radius is a compromise between the spatial representativeness of the COI and the inclusion of additional extreme events. We find that through the sampling process a significant increase in the rainfall amount of the top events could be reached. But we also believe that the general bias of the RAD-BC events would still remain, even we would enlarge the larger sampling radius. This is mainly due to the known underestimation of high intensity rainfall events in the radar.

8. Line 197 ff: In the independent fitting of distributions for different durations order relations problems may occur. This is accounted for in DWA (2012) by smoothing the parameters over the durations, which is a bit "old-fashion". Please, explain more in detail which method has been applied here and discuss also alternatives.

So far, we have not applied any smoothing across the different event durations. This was done to be in close agreement with the method used to generate the BW-Stat dataset. Also, the focus of the paper is more on the combined pooling process and bias-correction as well as the resulting changes in spatial patterns. But we briefly discuss the cross-duration relation and possible impacts on the varying spatial patterns of design rainfall events for different durations in the discussion section.

9. Line 217 ff: Please provide equations for the quantile mapping approach.

We included the respective equation in the revised version.

10. Fig. 3: Are the probability distributions compiled from all stations/locations together? If yes, how many stations are included?

They are calculated for all grid boxes. BW-Stat and RAD-BC have the same number of grid boxes since they are on the same grid. KOSTRA has a substantially lower number of grid boxes (hence the pdf is less smooth). We included this information in the figure caption.

11. Line 290: What is meant by spatial correlation analyses? Do you refer to correlations between rainfall and elevation? If yes, this is a cross-correlation but not a spatial correlation, which is usually used to quantify spatial persistence by correlation-distance relationships like the variogram, which by the why could have been employed for a more objective selection of the neighbourhood for sampling.

It is actually the cross correlation between the spatial patterns of REGNIE precipitation and the RAD-BC data. We changed this in the manuscript.

Regarding the variogram, we agree that this could also have been a valuable approach to identify a potential sampling area. However, also with this approach some subjective assumptions have to be made, since the correlation will be highest closest to the COI. But we want to sample in certain boundaries (linked to the size of convective cells). Based on the suggestions of Reviewer 1 we will move the figure describing the sampling statistics into the attachment. But we added some part on the uncertainty of the parameters underlying the sampling process in the revised version..

12. Fig. 5: Same question as for Fig. 3.

see reply above

13. Line 315: Why are you using an 80% confidence interval here; usually a 90% interval between 5% and 95% quantiles is used?

Thank you very much for pointing this out. We changed it to 5/95%.

14. Discussion/ conclusion: The new product has been compared against 2 reference data sets, but no strict validation has been carried out as usually desired. This is of course difficult since the truth is not known. However, often the long-term observations (>30 years) are applied as truth in a cross-validation mode. The application of this is also difficult here since the

RADKLIM data set itself is a merged product involving these stations which makes this infeasible. At least a discussion of this problematic is required and optimal would be some suggestions for further research.

Thanks for pointing this out. Since the methodological differences and the much lower spatial resolution of KOSTRA a one-to-one validation with KOSTRA is not possible. And the BW-Stat dataset is actually a pooled dataset itself.

Furthermore, we already know that both station based datasets have deficits in their spatial patterns caused by (i.) the limited number of stations included and (ii) the explicit consideration of the topography in the interpolation, which is (at least for short duration events) somehow questionable. We mention this problem of not having a clear validation dataset already in the manuscript, but we tried to better reflect it in the updated discussion of the results.

---

## Author Response (AR2)

**Review of „Enhancing the usability of weather radar data for the statistical analysis of extreme precipitation events"**

Dear Prof. Nadav Peleg, thank you very much for your kind coordination of the review process as well as for your suggestions to improve our manuscript.

As you requested, we deleted the text passages you marked in the manuscript as well as equation 7. We also added a reference for the statement in section 2.3.4 that heavy rainfall is often underestimated in the radar data and the appendix is now renamed to supplementary material and uploaded as a separate file.

We further included the recommendations of reviewer 2 in the updated version of the manuscript (see our reply on the next page).

On top of this we made some minor changes ourselves. These are as follows:
- We updated our affiliation to be in line with the respective university regulations.
- We added the word *ensemble* in the caption of Figure 6 so it reads now "… of the full (bootstrapping & sampling) 5000-member ensemble.)"
- We harmonized the font sizes in the figures 1,4 & 6 as well as of the figures in the supplementary material.
- We slightly changed the limits of the color scales in figure 6 in order to get rid of the decimals in the upper color bar and to harmonize the number of ticks.

Note that the changes we made for the figures are cosmetic changes only. In order to provide you an overview on what we changed in the text (we accepted the figure changes for better overview), we add the manuscript in track change mode at the end of this response.

Best regards,
Andreas Hänsler and Markus Weiler

**Review of „Enhancing the usability of weather radar data for the statistical analysis of extreme precipitation events"**

Dear Anonymous Reviewer,
thank you very much for your very detailed review of our manuscript and your very valuable suggestions to improve it.
Please find our response to the various points you raised below (in red).
Best regards,
Andreas Hänsler and Markus Weiler

The authors have taken most of my suggestions into account and substantially improved the manuscript. The descriptions of the methods and data sets used in this study are much clearer and easier to understand now. The method still seems to be rather complicated but the added flowchart helps to better follow the multiple steps of the process chain. They also improved the language and simplified some of the sentences. Therefore, I recommend accepting the manuscript for publication after addressing these few minor issues:

p.1, l.26-8: …various duration… --> …various durations…
The reviewer is right. We changed it accordingly.

p.6, l.180: So, in the end of the selection procedure 350 events from the five pixels are selected?
Yes, we indeed select 350 events. We made this more clear at this part of the manuscript.

p.9, l.265: What about a duration of 24 hours? There still seems to be an overestimation there, that cannot be explained by the lower spatial resolution of KOSTRA.
Thank you very much for pointing this out. Besides the spatial resolution the methodological differences are the major source of differences between KOSTRA and the other datasets. We made this more clear at this point of the manuscript.

p.12, l.339: …that seems a to be… --> …that seems to be…
The reviewer is right. We changed it accordingly.

p.14, l.421: …that are might not affected… --> …that are not affected…
The reviewer is right. We changed it accordingly.

p.19, Fig.2: Step 8: Are the location, shape and scale parameter really taken from step 4, as it is written in the flowchart, or are they taken from step 5?
The reviewer is right. We changed it to step 5.

[revised manuscript text omitted]